# Impact of uncertainty maps on manual editing of rectal cancer segmentation in radiotherapy

Federica C. Maruccio[1]  ORCID                           F.MARUCCIO@NKI.NL
Rita Simões[1]                                                R.SIMOES@NKI.NL
Fokie Cnossen[2]                                            F.CNOSSEN@RUG.NL
Christian Jamtheim Gustafsson[3,4]       CHRISTIAN.JAMTHEIMGUSTAFSSON@SKANE.SE
Sanne Conijn[1]                                              S.CONIJN@NKI.NL
Alice Couwenberg[1]                                   A.COUWENBERG@NKI.NL
Suzan Gerrets-van Noord[1]                         S.GERRETS@NKI.NL
Inge de Jong[1]                                             I.D.JONG@NKI.NL
Vivian van Pelt[1]                                          V.V.PELT@NKI.NL
Lisa Wiersema[1]                                        L.WIERSEMA@NKI.NL
Joëlle E. van Aalst[5]                               J.E.VAN.AALST@UMCG.NL
Jan-Jakob Sonke[1]                                        J.SONKE@NKI.NL
Charlotte L. Brouwer[5]                            C.L.BROUWER@UMCG.NL
Tomas M. Janssen[1]                                     T.JANSSEN@NKI.NL

[1] *Department of Radiation Oncology, The Netherlands Cancer Institute, Amsterdam, the Netherlands*
[2] *Department of Artificial Intelligence, Bernoulli Institute of Mathematics, Groningen, Netherlands*
[3] *Department of Hematology, Oncology, and Radiation Physics, Skåne University Hospital, Lund, Sweden*
[4] *Department of Translational Medicine, Lund University, Malmö, Sweden*
[5] *Department of Radiation Oncology, University Medical Center Groningen, Groningen, Netherlands*

**Editors:** Accepted for publication at MIDL 2026

## Abstract

Uncertainty maps provide a quantitative and visual representation of the estimated confidence of Deep Learning (DL) models in contouring predictions and have been proposed to improve clinicians' efficiency during manual review. However, uncertainty maps are not currently integrated into clinical workflows, and evidence on their actual benefit in clinical decision-making remains limited. This study investigates the impact of simulated uncertainty maps on clinicians' behaviour during manual editing of high-quality clinical target volume (CTV) contours in rectal cancer radiotherapy. An inter-observer variability dataset of ten patients was used to simulate meaningful DL uncertainty maps and contours. Six clinicians edited the contours across two editing sessions, with and without uncertainty maps. For each session, editing time, editing amount, questionnaire responses, and interview feedback were collected to assess the impact both quantitatively and qualitatively. Editing time and editing amount were comparable with and without uncertainty maps, while both measures decreased significantly in the second editing session, indicating a learning effect from task repetition. Qualitative feedback showed that clinicians' decisions were shaped more by human factors, such as workload, mood, memory and anchoring biases, than by the uncertainty maps. Moreover, the study revealed low clinician trust in the uncertainty maps, which were used primarily for confirmation rather than decision-making. The findings suggest that the value of uncertainty maps may be limited for high-quality contours and highlight the need to investigate their relevance for different use cases.
Code is available at https://github.com/NKI-RT/contour-uncertainty-impact.git.

**Keywords:** Segmentation, manual editing, uncertainty maps, deep-learning, radiotherapy

## 1. Introduction

In radiotherapy (RT) workflow, delineating targets and organs at risk (OARs) for treatment planning is a highly time-consuming task. This challenge is even more pronounced in online adaptive RT, where the dose distribution of each fraction is adapted to the patient's daily anatomy. In this setting, targets and OARs need to be re-delineated for every treatment fraction under substantial time pressure, as the patient is waiting on the table for treatment delivery. Over the past decade, the integration of deep learning (DL)-based auto-segmentation into RT workflows has been growing, aiming to automate delineation. DL models have demonstrated potential to reduce contouring time and inter-observer variability (IOV), thereby enabling faster and more consistent delineations (Vinod et al., 2016). However, DL contours can be prone to clinically relevant errors, making manual review and correction by radiation oncologists (ROs) and radiation therapists (RTTs) an essential part of the clinical workflow. Inaccurate auto-segmentation can directly affect patient outcomes, compromising target coverage or increasing the risk of radiation-induced toxicity. Moreover, the need for manual adjustments diminishes time savings and can undermine clinicians' trust in DL tools.

Uncertainty estimation has emerged as a promising strategy to support manual editing by providing information on the model's confidence in its predictions (De Biase et al., 2024b). Such information can be visualised as an uncertainty map, highlighting the regions that may be unreliable due to different sources of uncertainty. Specifically, two types of uncertainty can be distinguished. Epistemic uncertainty reflects limitations in the model itself, for example, due to insufficient or unrepresentative training data. Aleatoric uncertainty captures noise or ambiguity inherent in the data, such as low image quality and IOV in the manual delineations of the training data (Kendall and Gal, 2017). When presented alongside the DL contours, uncertainty maps may guide clinicians toward regions requiring closer inspection, potentially reducing variability and improving efficiency (Begoli et al., 2019; Kompa et al., 2021; Ren et al., 2024; van Aalst et al., 2025).

Several studies have focused on quantifying DL-based uncertainty in the context of RT delineations across various tumour sites and employing different approaches (Balagopal et al., 2021; Min et al., 2023; Outeiral et al., 2023; Sahlsten et al., 2024; Maruccio et al., 2025; van Aalst et al., 2025). Among these, Monte Carlo dropout (Gal and Ghahramani, 2016) and deep ensemble modelling (Lakshminarayanan et al., 2017) are the most widely used methods, introducing stochasticity into the model during inference and quantifying it through uncertainty metrics, usually predictive entropy or variance (Wahid et al., 2024; van Aalst et al., 2025). In parallel, studies have explored how this information can be effectively visualised and communicated to clinicians (Huet-Dastarac et al., 2024; De Biase et al., 2024b). Despite this progress and several works highlighting the need to investigate the impact of uncertainty maps in clinical workflows (van Rooij et al., 2021; Maruccio et al., 2024; Wahid et al., 2024; De Biase et al., 2024a), this topic remains largely unexplored. To date, only a single study has assessed the influence of DL-based uncertainty maps on clinicians' editing behaviour (Rogowski et al., 2025).

While major errors are usually easy to detect and correct, the key question is whether uncertainty maps can help clinicians identify smaller yet clinically relevant inaccuracies that might otherwise be overlooked. It also remains unclear whether presenting uncertainty in-

formation can increase efficiency by helping clinicians navigate more quickly through regions of low uncertainty. More information does not necessarily translate to improved efficiency, as additional visual cues may lead to information overload and even reduce performance (Huet-Dastarac et al., 2024).

Rogowski et al. (2025) concluded that uncertainty information can influence clinicians' decision-making and reduce segmentation time. However, their findings may be biased by the specific performance of their model, as no reliability analysis was performed on the uncertainty estimates (Guo et al., 2017). The usefulness of uncertainty maps depends on the accuracy and calibration of the underlying uncertainty estimation model. If the model is poorly calibrated, the resulting maps may be misleading (Rabe et al., 2025). This explains the difficulty in designing experiments that meaningfully isolate the effect of uncertainty information on clinicians' editing behaviour, as a ground-truth for uncertainty does not exist.

To address this, we used manual contours as a proxy for high-quality DL predictions and IOV maps as a surrogate for uncertainty maps with a clear and clinically interpretable meaning. Notably, IOV has previously been used as a form of ground truth uncertainty (Li et al., 2023). This choice also aligns with expected developments in the field. As segmentation models continue to improve, their output is likely to approach, on average, the quality of manual delineations. With larger training datasets and improved imaging quality, model- and imaging-related uncertainty may diminish, while IOV will remain a fundamental source of uncertainty (Wahid et al., 2024). However, even with high-quality predictions, clinicians will still need to manually review the predictions, as failures can happen.

The current work investigates whether presenting meaningful uncertainty information affects clinicians' editing behaviour during manual editing of clinical target volume (CTV) high-quality contours in rectal cancer RT. We evaluated this through a two-session study involving six clinicians, analysing editing time, editing amount, inter-observer variability, and qualitative feedback.

**Contributions.** The key contributions of this work are the following:

- A robust study design was implemented to isolate the impact of uncertainty maps on clinicians' editing behaviour from the performance of any specific DL model, by using high-quality manual contours and uncertainty derived from IOV as a proxy of accurate DL outcomes. The design accounted for order effects, memory bias and confounders such as workload and learning effect. Quantitative metrics, including editing time and geometric measures, were combined with qualitative feedback from questionnaires and interviews to provide a comprehensive assessment of clinicians' editing behaviour.

- It was found that, under conditions of high-quality initial contours and uncertainty maps, uncertainty information did not significantly affect editing time, amount of editing, or inter-observer variability. Instead, human factors such as mindset, workload, and learning effect from task repetition played a prominent role in shaping editing behaviour.

- The study revealed a low trust of the clinicians in the uncertainty maps, which were used primarily for confirmation rather than decision-making. The findings also identify scenarios where uncertainty information may hold greater value, e.g., lower-quality contours or more complex anatomies.

## 2. Methods

### 2.1. Data set

We retrospectively retrieved a dataset of ten patients (IRB approval IRBd24-340), treated for intermediate-risk or locally advanced rectal cancer using the Unity MR-Linac (Elekta AB, Stockholm) at the Netherlands Cancer Institute (NKI), Amsterdam, The Netherlands. Three-dimensional (3D) T2-weighted MRI images were acquired with a field of view of $400 \times 448 \times 249 \, \text{mm}^3$, repetition time (TR) of 1300 ms, and echo time (TE) of 128 ms. The dataset includes 10 MRI scans with a voxel size of $0.57 \times 0.57 \times 1.20 \, \text{mm}^3$.

### 2.2. Generation of uncertainty maps

For a prior evaluation of inter-observer variability (IOV) (Silvério et al., 2024), five experts in RT rectal cancer delineation, including a radiation oncologist (RadOnc) and four radiation technology therapists (RTTs) trained for MR Linac rectal cancer treatment, manually contoured the mesorectum clinical target volume (CTV) on all MRI scans according to national guidelines (Valentini et al., 2016). To minimise the influence of external factors, such as model performance and model calibration, on our primary research question, a single manual contour was randomly selected from the five experts' contours available for each patient in the IOV dataset as a proxy for a high-quality "DL contour".
For each patient, the local standard deviation of the distances between the selected contour and the remaining four manual contours in the IOV dataset was computed, using it as a proxy for a meaningful "DL uncertainty map". This setup enabled us to isolate and evaluate the effect of providing clinically relevant uncertainty information, capturing primarily aleatoric uncertainty. For simplicity, throughout the remainder of this manuscript, the selected contours and the derived uncertainty maps for the 10 patients are referred simply to as *contours* and *uncertainty maps*, respectively.
The uncertainty maps were post-processed to optimise their visualisation within Mirada RTx (Mirada Medical Ltd, Oxford, UK), one of the delineation software packages used in clinical practice. A Gaussian filter was first applied to smooth local intensity fluctuations, after which the maps were normalised to a 256-level colour scale by dividing by the maximum uncertainty value across all patients and scaling to 256 to ensure consistent interpretation of intensity values. To enhance visibility when overlaid on the contour, each map was dilated by 2 pixels in-plane and intensity values were clipped to the 1st-99th percentile range. The resulting uncertainty volumes were exported as positron emission tomography (PET)-encoded DICOM images, to enable their overlay on the corresponding MRI scans (Figure 1) (Rogowski et al., 2025).

### 2.3. User study

Ten clinicians from the NKI hospital were invited to participate in the study; six agreed to take part, including one RadOnc specialised in rectal cancer treatment planning delineations and five RTTs trained to delineate in MR-guided OART workflows. Their clinical experience in RT ranged from 6 to 21 years, while their experience with rectum CTV delineation ranged from less than 1 year to 5 years (see Table 1 for details).
Clinicians participated in two contour-editing sessions, separated by a two-month interval

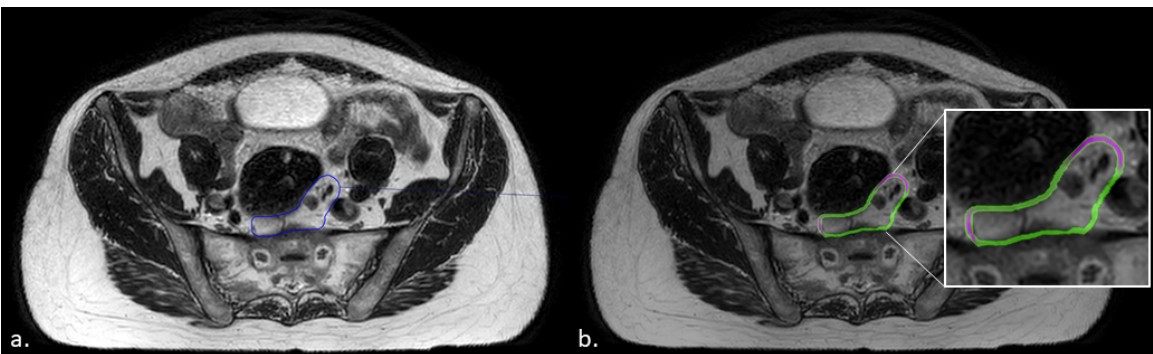

Figure 1: Example of the contour and uncertainty map visualisation in Mirada. a) MR image of a representative patient in an axial view with the CTV contour shown in blue. b) Corresponding uncertainty map displayed using consistent settings across all participants (green-magenta colourmap, 30% transparency, and window-level between the minimum and 75% of the full range).

Table 1: Information on the participants in the study.

| Participant ID | Job title | Clinical experience [years] | Automatic segmentation experience [years] | Rectum CTV delineation experience [years] | Experience with Mirada [Yes/No] |
|---|---|---|---|---|---|
| A | RadOnc | 6 | 0 | 4 | Yes |
| B | RTT | 13 | 1.5 | 3 | Yes |
| C | RTT | 20 | 5 | 5 | Yes |
| D | RTT | 21 | 1.5 | 2.5 | No |
| E | RTT | 14 | < 1 | < 1 | Yes |
| F | RTT | 19 | 5 | 3 | Yes |

(Haygood et al., 2018). In both sessions, they were requested to review and, if they wished, edit the contours as in an adaptive workflow under two conditions: *without* and *with* uncertainty maps. To isolate the effect of uncertainty maps, the 10 patient cases were split into two groups stratified by IOV. Each group was presented without uncertainty maps in one session and with uncertainty maps in the other. Therefore, in a counterbalanced within-subject design, all participants edited 10 cases in each session, 5 without uncertainty maps, followed by 5 with uncertainty maps, as illustrated in Figure 2. Within each group, the case order was independently randomised for each participant (Appendix C). This design ensured that each participant edited every patient twice, once with and once without uncertainty information. Participants were informed that both the contours and the uncertainty maps were generated by a DL model.

One month after completion of the second session, one-on-one semi-structured interviews were conducted to obtain additional insights into the use and perceived value of the uncertainty maps.

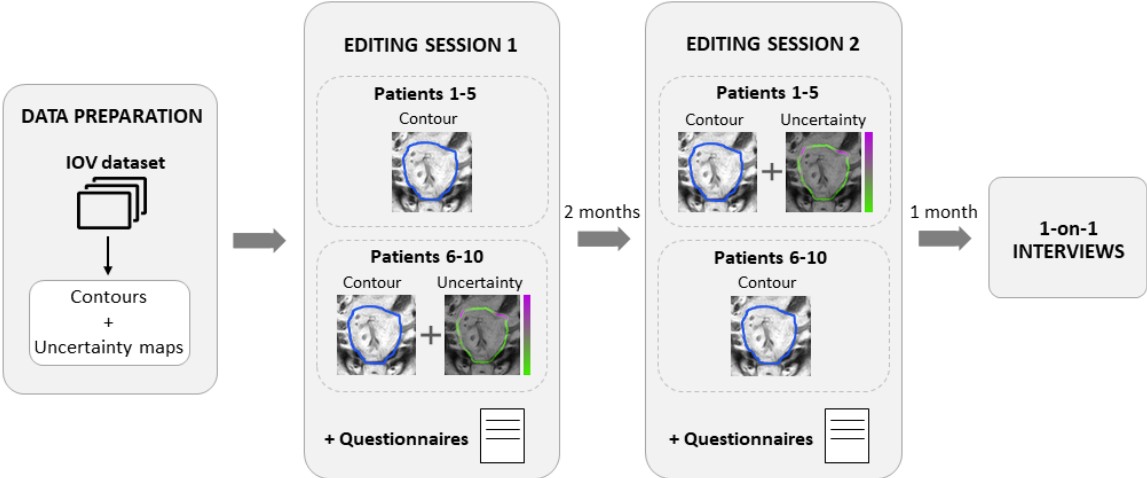

Figure 2: Study design. Independent manual contours from an inter-observer variability (IOV) dataset of ten patients were used to simulate high-quality DL contours and uncertainty maps. Participants edited the contours in two sessions, alternating between conditions with and without uncertainty maps, and completed questionnaires on user experience. One-on-one semi-structured interviews were then conducted.

### 2.3.1. PREPARATION EXPERIMENTS

During a preliminary meeting, participants were briefed on the study and its design. Moreover, they were provided with an explanation of the uncertainty maps, followed by a discussion on alternative visualisation strategies. They were then asked to give their consent for participation in the research, the processing of personal data, and the use of screen recordings by signing an informed consent form (Appendix A). Based on the conversation with the participants and findings of Huet-Dastarac et al. (2024), which highlighted clinicians' preference for voxel-level binary uncertainty information, a colour-blind safe green-magenta colour map was implemented in Mirada DBx for visualisation.

Prior to the main experiment, a pilot session was conducted in which participants received a first version of the information sheet. This document outlined all the steps to be followed during each session, including instructions on how to access the data, record their screen using Microsoft Stream (Microsoft Corporation, Redmond, WA, USA), visualise the uncertainty maps to ensure consistent settings across all users, including standardised colorbar configuration, window-level parameters, and transparency, and save their sessions.

Feedback from the pilot session was used to refine the information sheet and uncertainty visualisation (Appendix B). The experimental data were then uploaded to Mirada DBx to begin the main study.

### 2.3.2. EDITING SESSIONS

Clinicians were instructed to follow the provided information sheet during each editing session, review and adjust the contours if needed as in an adaptive workflow, record their

screen for the full duration of the session, and complete a questionnaire afterwards. In both sessions, they rated the quality of the initial contour (question Q1, scale 1-4) and their confidence in the final edited contour (question Q2, scale 1-5). After completing session 2, clinicians filled out an additional user-experience questionnaire containing 9 statements on a 7-point Likert scale ranging from 1 (strongly disagree) to 5 (strongly agree) comparing the two conditions, contour only and contour with the uncertainty map, similar to De Biase et al. (2024b). The full questionnaires are included in Appendix E.3.

### 2.3.3. Debriefing interviews

One-on-one semi-structured debriefing interviews were conducted in the final session of the study to support and contextualise the quantitative findings. Each interview lasted approximately one hour and followed a five-stage structure. First, participants were introduced to the purpose and format of the interview and were asked to provide consent for audio recording. Second, three representative cases previously edited by the participant during both editing sessions were reviewed using their corresponding screen recordings as a memory cue (Dercksen et al., 2024). Cases were selected by visualising the 3D edits from both sessions and examining histograms of edited voxels across uncertainty bins (Rogowski et al., 2025), with the aim of capturing a range of editing scenarios. Participants were invited to describe their reasoning and decision-making processes, and were asked case-specific questions to clarify these processes. Third, general questions regarding user experience were discussed, informed by the participants' questionnaire responses. Next, preliminary study results, including editing time, geometric analysis metrics, and questionnaire ratings for all participants, were presented using a PowerPoint presentation. Participants were then asked to reflect on these results and provide their interpretation based on their own experience. Finally, participants were debriefed and informed that the contours and uncertainty maps used in the study had not been generated by a DL model, and the rationale for this was explained. Participants were asked to keep this information confidential until all interviews were completed and were invited to comment on this aspect of the study.

## 2.4. Analysis

To evaluate the impact of uncertainty maps on the contour correction process, a combination of quantitative and qualitative methods was used. Editing times, geometric metrics, and inter-observer variability (IOV) were analysed to quantify differences between the tested conditions (without and with uncertainty maps) and between editing sessions. For all statistical comparisons, a Wilcoxon signed-rank test with Bonferroni correction was performed using the Python SciPy package (Virtanen et al., 2020). In addition, paired effect size analyses using Cohen's d were conducted to better contextualise the results, particularly in the presence of a limited sample size. The quantitative findings were further contextualised through questionnaires and interview transcripts, enabling a comprehensive interpretation of participants' interactions with the uncertainty maps.

### 2.4.1. Editing time

Editing time for each case was extracted from participants' screen recordings and compared across conditions and editing sessions to assess the potential impact of the uncertainty maps

and examine session-related effects. The analysis was performed on the participant-level, on the patient-level and by pooling all data together.

### 2.4.2. Geometric metrics

To evaluate differences in editing magnitude between conditions and across sessions, the contours provided to the participants were treated as reference contours. For each patient and participant, the Dice score and added path length (APL) (0 mm threshold) (Vaassen et al., 2020) metrics were computed between the reference and the edited contours. Higher Dice scores and lower APL values indicate greater geometric similarity between the reference and edited contours, reflecting minimal editing performed by the participants.

### 2.4.3. Inter-observer variability

Inter-observer variability (IOV) within each condition and editing session was quantified using the mean pairwise conformity index (CI) and compared to the initial manual IOV dataset (baseline) for each patient. Additionally, 3D IOV maps were generated for each patient and condition by calculating the local standard deviation of distances between each participant's contour and the corresponding median contour. These maps were used qualitatively to visualise the spatial distribution of the variability and verify consistency with the CI results.

### 2.4.4. Questionnaires and interviews

Audio recordings of the interviews were transcribed verbatim using Spinach AI (StayIn, Nashville, TN). The interview transcripts and the questionnaire responses were reviewed and analysed to identify recurring themes and insights. The analysis was conducted in collaboration with a cognitive psychologist specialised in human-machine interactions (F.C.) to ensure interpretation of participant responses was consistent with established principles of experimental psychology. Key observations focused on the perceptions and experiences of the participants with the uncertainty maps and were used to interpret the quantitative findings from the editing sessions.

## 3. Results

### 3.1. Editing time

Median editing time per patient was $4.2 \pm 3.3$ min when editing without uncertainty maps and $4.1 \pm 3.3$ min when editing with uncertainty maps, showing no significant difference between the two conditions ($p = 0.57$; $|d| = 0.01$) However, editing time was significantly shorter in session 2 ($3.4 \pm 2.1$ min) compared to session 1 ($6.2 \pm 3.4$ min) ($p = 1.63 \times 10^{-7}$; $|d| = 0.87$; Figure 3). Detailed participant- and patient-level timings are provided in Appendix E.1.

### 3.2. Geometric metrics

Similar to the editing time results, geometric metrics exhibited comparable outcomes with and without the use of uncertainty maps. Specifically, the median Dice score between the

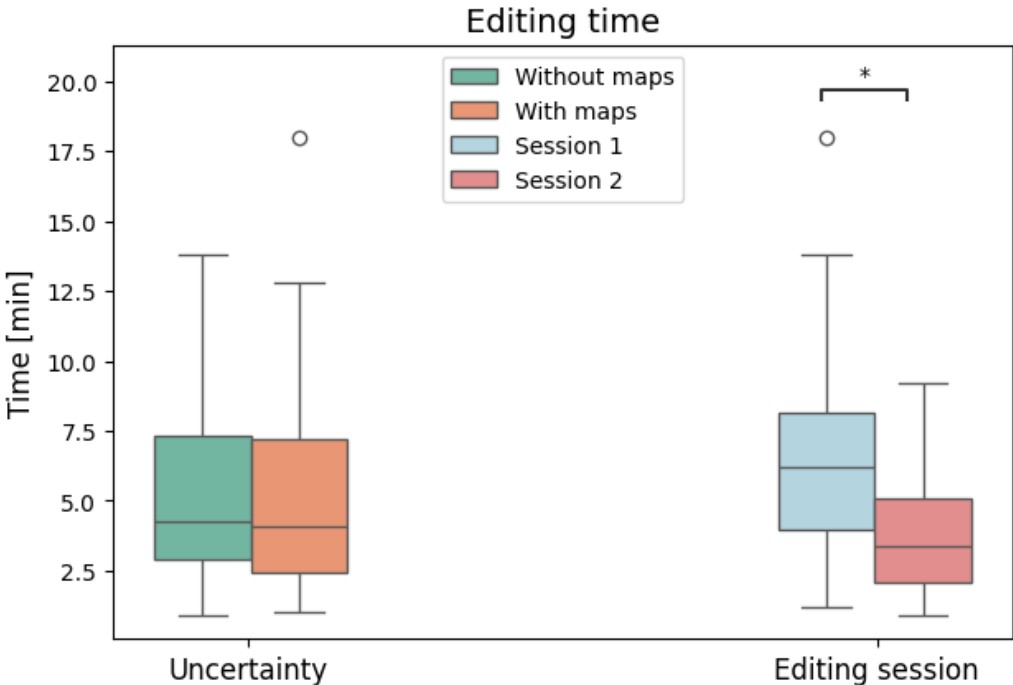

Figure 3: Impact of uncertainty maps and sessions on editing time. Boxplots show distributions of editing time (minutes) across all participants and patients for the two tested conditions (without vs. with uncertainty maps) and for the two editing sessions (Session 1 vs. Session 2). Statistical significance was assessed using the Wilcoxon signed-rank test, and * indicates $p < 0.01$.

initial contours and those edited without access to uncertainty maps was $0.99\pm0.01$, equivalent to the median Dice score for contours edited with uncertainty maps ($p = 0.82; |d| < 0.1$). In contrast, comparing across sessions revealed a significant difference in editing amount, with the Dice score between the given contours and those edited in session 2 ($0.99 \pm 0.01$) exceeding that for session 1 ($0.98 \pm 0.02$) ($p = 4.66 \times 10^{-5}; |d| = 0.34$; Figure 4a).

A similar trend was observed for the APL metric. No significant differences were observed between the conditions without uncertainty maps ($2.5\pm3.4\,\mathrm{cm}^3$) and with uncertainty maps ($1.7 \pm 3.8\,\mathrm{cm}^3$) ($p = 0.56; |d| < 0.1$). However, median APL in session 2 ($1.1 \pm 3.5\,\mathrm{cm}^3$) was significantly lower than in session 1 ($3.0 \pm 3.6\,\mathrm{cm}^3$), indicating that participants performed less editing in the second session, independent of the presence of uncertainty maps ($p = 7.66 \times 10^{-5}; |d| = 0.44$; Figure 4b). Detailed participant- and patient-level results for both metrics are provided in Appendix E.2.

### 3.3. Inter-observer variability

No effect of providing uncertainty maps was observed on IOV. As shown in Figure 5, the median conformity index (CI) was comparable between conditions with and without uncertainty maps ($0.97 \pm 0.02$) ($p = 1.00; |d| < 0.1$). Session-wise comparisons also revealed

Figure 4: Impact of uncertainty maps and sessions on editing amount. Boxplots show distributions of Dice score and APL values between initial contours and edited contours across all participants and patients for the two tested conditions (without vs. with uncertainty maps) and for the two editing sessions (Session 1 vs. Session 2). Higher Dice scores and lower APL values indicate greater geometric similarity between the reference and edited contours, reflecting less editing performed by the participants. * indicates $p < 0.01$.

no statistically significant difference between session 1 and session 2 ($p = 0.26; |d| = 0.55$). However, median CI increased substantially from the baseline IOV dataset ($0.83 \pm 0.04$) to session 1 ($0.96 \pm 0.02$) ($p = 7.82 \times 10^{-3}; |d| = 3.5$) and session 2 ($0.98 \pm 0.02$) ($p = 7.82 \times 10^{-3}; |d| = 3.5$), indicating higher agreement between participants when editing from a provided starting contour compared to fully manual segmentations from scratch. Figure 6 provides an illustrative example of IOV maps for a representative patient. In these maps, brighter regions correspond to higher variability between observers, while darker regions indicate higher conformity. Baseline contours exhibit larger areas of variability, whereas the IOV maps for sessions 1 and 2 are similar and show reduced variability, consistent with the CI results.

## 3.4. Questionnaires and interviews

On a participant-level, questionnaire ratings on the perceived quality of the initial contour (Q1) and on confidence in the final decision (Q2) were equivalent or higher in the second session compared to the first (see Tables 3 - 4 and Figure 10 in Appendix). Such a trend was not observed between the tested conditions. Across subquestions, the contour-only condition generally received higher ratings than the contour+uncertainty condition. Compared with contour-only, participants rated the presence of uncertainty maps as less helpful (4/7

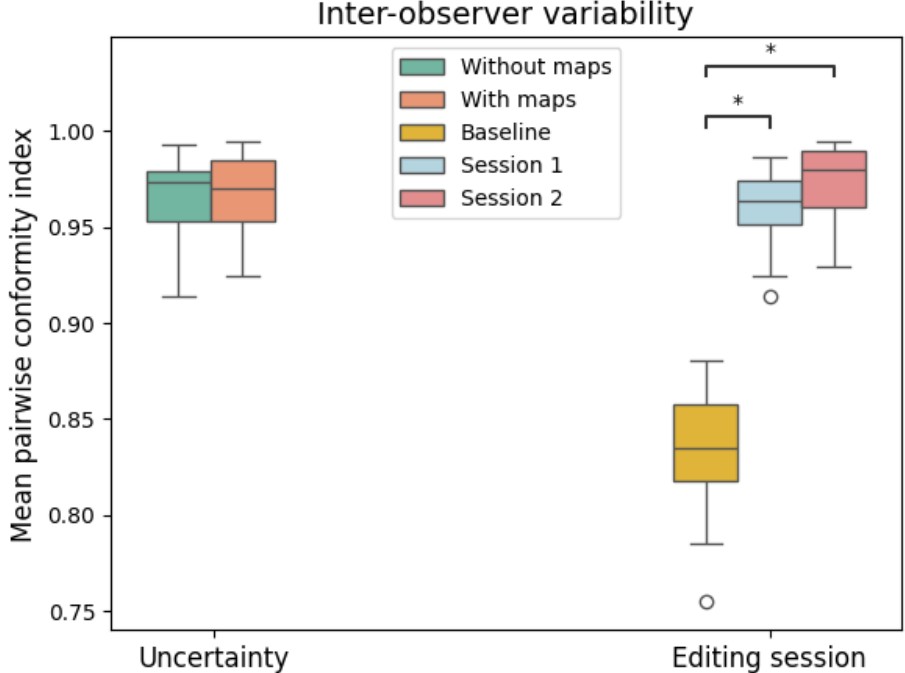

Figure 5: Impact of uncertainty maps and sessions on inter-observer variability (IOV). Box-plots display distributions of conformity index (CI) values across all participants and patients for the two tested conditions (without vs. with uncertainty maps) and for the two editing sessions (1 vs. 2). CI values for the baseline IOV dataset are also shown for comparison. Higher CI values indicate greater conformity among observers' contours, reflecting lower IOV. * indicates $p < 0.01$.

vs 6/7), less time-saving (3/7 vs 7/7), and less feasible (4/7 vs 6/7) for clinical use. Moreover, the two statements on preferences revealed a higher preference for using contour-only over delineating from scratch ($>6.5/7$) than using contour+uncertainty over contour-only ($<3.5/7$). Responses on complexity, confusion, influence on decision-making and increase in confidence were similar (see Table 5 and Figure 10 in Appendix).

Interview responses revealed several recurring themes. First, most participants emphasised the high quality of the provided initial contours, noting that the majority of edits were minor. Participants experienced a learning curve due to no familiarity with the Mirada software (participant D), editing automatic contours instead of delineating from scratch (participant A), limited experience with rectum cancer patients (participant E), the absence of typical clinical reference information (all five RTT participants), and adapting to study procedures (all participants). Additionally, three participants (A, C and D) reported an increased workload in the clinic during session 2. All participants revealed a shift from a more critical and perfectionist approach in session 1 to a more pragmatic mindset in session 2 due to prior familiarity with the study setup and contour quality. Furthermore, four participants (A, C, E, and F) reported that, despite the two-month interval between editing sessions, they recognised the anatomical details of one or more patients in session 2. Some participants found the need to toggle the uncertainty maps on and off, mainly to

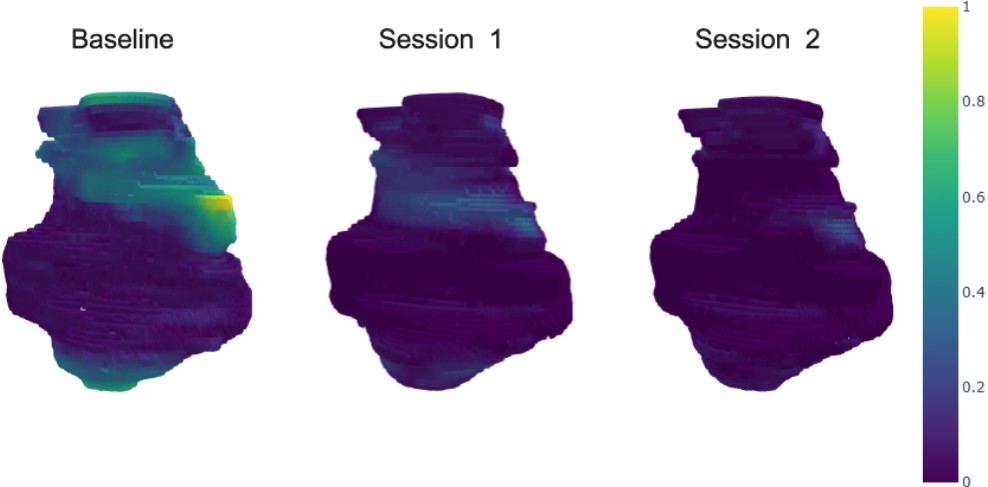

Figure 6: Example of inter-observer variability (IOV) maps. IOV maps for a representative patient are displayed for the baseline IOV dataset and editing sessions 1 and 2. Brighter regions correspond to higher variability between observers, while darker regions indicate higher conformity.

better view the underlying anatomy, distracting. Finally, several participants reported low trust in the uncertainty maps, using them primarily as confirmation for their own initial assessment rather than as a decision-support tool to change their assessment. A report of all interviews is provided in the Appendix E.4.

## 4. Discussion

This study investigated the impact of uncertainty information on clinicians' behaviour during manual editing of CTV rectal contours. Using a robust experimental design, independent of any DL model performance and controlling for order effects, memory bias and other human confounders, it was found that uncertainty maps did not significantly affect clinical behaviour, in terms of editing time, editing amount and inter-observer variability. Human factors such as mindset, workload, and learning effect were identified as the main drivers in shaping clinician editing behaviour, as changes occurred between editing sessions.

Editing time and editing amount did not differ significantly between the two tested conditions, without and with uncertainty maps, contrary to what was hypothesised or observed in previous studies (De Biase et al., 2024b; Rogowski et al., 2025; van Aalst et al., 2025; Maruccio et al., 2024; Huet-Dastarac et al., 2024). Several factors help explain why the uncertainty maps had no measurable effect. First, their practical use during the task was limited. Screen recordings revealed that participants typically inspected the maps only briefly at the beginning, mainly in the sagittal view to assess the overall uncertainty, and then disabled them for most of the editing process. Participants reported that the overlay reduced anatomical visibility and that the thickness of the map near the contour borders hindered precise adjustments. The consequent need to toggle the map on and off was perceived as disrupting their workflow, which may have discouraged consistent use and contributed to

lower questionnaire ratings of feasibility and time savings. Second, limited trust in the maps further reduced their influence. Participants tended to rely on their own judgment when the uncertainty information conflicted with their assessment, using the maps primarily as a tool for confirmation to justify their actions rather than decision-support, consistent with well-known confirmation bias patterns (Shafir et al., 1993). Several interviewees indicated that trust would need to be built through repeated exposure to reliable and clinically meaningful uncertainty information, as stated also by Kompa et al. (2021). Without such trust, even well-designed visualisations may have minimal behavioural impact.

Although no differences were observed between the conditions, the editing time was significantly influenced by the editing session itself. Two main factors emerged from the interviews which help explain this finding: learning effect and memory bias. Participants A and E showed the largest reductions in editing time between sessions (Appendix E.1), suggesting a pronounced learning effect (Tao et al., 2019). Specifically, participant A initially struggled to adopt an online-adaptive mindset, being accustomed to treatment-planning workflows where delineations are created from scratch without strict time constraints. After recognising the generally good quality of the initial contours, their editing approach shifted toward performing only clinically relevant edits. Participant E had only recently begun delineating rectum cases, and their experience likely increased substantially during the two-month interval working in the clinic, contributing to faster editing in the second session. For the remaining participants, learning effects were present but less pronounced. Contributing factors included lack of experience with Mirada (participant D), absence of additional clinical information (all RTT participants) (Chaves-de Plaza et al., 2022), and repetition of the experimental tasks (Tao et al., 2019). Overall, these factors likely produced learning effects that outweighed a potential intervention effect (uncertainty map usage). Moreover, even if memory-mitigating steps were taken according to the literature (Haygood et al., 2018), such as time gap of 2 months between sessions and presentation of the cases in different and random orders across sessions and participants, participants reported remembering the anatomy and contours for certain cases. This memory bias may have facilitated more rapid decision-making during the second session, contributing further to the observed reduction in editing time.

Furthermore, the amount of editing performed in the second session was significantly lower than in the first session, in line with the observed reduction in editing time. Interview responses highlighted two distinct editing mindsets across the study sessions. In session 1, participants tended to adopt a more critical and perfectionistic approach, whereas in session 2, they reported being more pragmatic in their edits. Additionally, a few interviewees mentioned being aware that changes in mood, within the same day or across different days, can affect their delineation behaviour, both during the study and in routine clinical practice. Workload emerged as another important factor influencing editing behaviour. When clinicians are tired or under increased pressure, they may be more inclined to accept imperfections, effectively raising their error tolerance. A few participants reported experiencing higher clinical workload during the period of session 2, which may have led to fatigue and contributed to more superficial edits and, consequently, shorter editing times in the second session. These observations align with well-established human factors known to shape clinician performance and contribute to IOV in contouring (Cowen et al., 2025; Das et al., 2021).

IOV analysis revealed no significant differences between the two tested conditions or between the two sessions, contrary to what suggested in the literature (van Aalst et al., 2025). However, a substantial reduction in IOV was observed when comparing the initial IOV dataset with the edited contours in both sessions. This reduction is likely attributable to anchoring bias, whereby participants were influenced by the provided initial contour (Mackay et al.). Anchoring bias may also explain the higher perceived contour quality reported in the questionnaires for session 2, despite the contours being identical across sessions. Over time, clinicians may become more willing to accept and rely on the initial information, which is consistent with the higher confidence in final decision reported for session 2.

Our findings differ from those of Rogowski et al. (2025), who observed an influence of the uncertainty maps on clinicians' decision-making, quality perception, and confidence in the DL contours. A possible explanation lies in differences in study design, as their setup did not control for the learning effect from task repetition since the tested conditions were not randomised across sessions. In contrast, our design explicitly accounted for such confounders. However, other sources of learning effect were less present in their work as all participants oncologists were already experienced with workflows similar to the experimental task. Additionally, we enforced fixed visualisation settings to ensure consistency across participants, whereas Rogowski et al. (2025) allowed oncologists to adjust the blending between the MR image and the uncertainty map, potentially enhancing usability.

Importantly, the absence of significant differences should not be interpreted as evidence that uncertainty maps are generally ineffective, but rather that their added value may be constrained when contour quality is already near expert-level. This study was designed as an exploratory evaluation in an idealised scenario, to isolate the effect of uncertainty information itself and establish a controlled experimental framework. Within this context, the results suggest that, beyond a certain level of technical performance, clinician behaviour may be influenced more by cognitive, behavioural, and workflow-related factors than by additional information. Despite the non-significant impact, participants noted that uncertainty maps may still be valuable in more difficult cases or when DL contours require substantial corrections. Interestingly, RTT participants considered uncertainty information to be more useful in offline settings, where it could support RadOncs in achieving greater consistency. Conversely, the RadOnc participant expressed openness to using additional information but considered uncertainty maps as potentially more beneficial in an online adaptive context, where RTTs could use them to save time during fast-paced workflows. The framework introduced in this work, therefore, provides a basis for future prospective studies using DL-based uncertainty in clinically realistic settings, where contour quality, uncertainty sources and reliability, and time pressure may interact differently and where uncertainty maps may have a more pronounced practical impact.

This study has several limitations. First, the dataset included only ten patients, the number of participating clinicians was limited to six, and the investigation focused on a single tumour site and on target delineation, which may limit the generalisability of the findings. However, the cohort size and number of evaluated tasks are consistent with prior IOV and clinician observer studies in RT, where datasets typically include a small number of patients and expert delineations due to the substantial annotation burden (Guzene et al., 2023). Furthermore, while the participant group included five RTTs, only one RadOnc took part in the study, limiting the representativeness of that role. Although RTTs routinely perform

contour editing and online adaptation in Dutch clinical practice and are therefore end users of AI-assisted delineation tools (van Pelt et al., 2021; Rasing et al., 2022; Goudschaal et al., 2025), professional role may influence the interpretation and the use of uncertainty information. Future studies with larger and more diverse cohorts are therefore needed. In addition, four participants had contributed to the IOV dataset approximately two years prior to the study; however, the long time interval makes it unlikely that this introduced memory bias.

## 5. Conclusions

Uncertainty maps for high-quality contours did not significantly influence editing behaviour, whereas human factors had a more pronounced impact. More research is needed to determine whether there is a "sweet spot" where uncertainty maps offer real value, between cases with obvious errors and those with already high-quality contours, for which this study showed limited impact. Moreover, human factors, such as trust and usability, play a crucial role in determining whether a tool can be effectively translated into clinical practice. Future research should therefore prioritise user-centred design, systematically assess the value of uncertainty maps across use cases, and conduct longer-term prospective studies in which learning effects stabilise, trust can develop, and the true impact of uncertainty maps on clinical practice can be rigorously assessed.

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

## Appendix A. Informed consent form

# Informed Consent Form

### Impact of uncertainty information on correction process

- I have read the information about the research. I have had enough opportunity to ask questions about it.

- I understand what the research is about, what is being asked of me, which consequences participation can have, how my data will be handled, and what my rights as a participant are.

- I understand that participation in the research is voluntary. I myself choose to participate. I can stop participating at any moment. If I stop, I do not need to explain why. Stopping will have no negative consequences for me.

- Below I indicate what I am consenting to.

**Consent to participate in the research, to the processing of my personal data and to the use of my screen recordings:**

☐ Yes, I consent; this consent is valid until 31-12-2025. I know that until 31-12-2025 I can ask to have my data withdrawn and erased. I can also ask for this if I decide to stop participating in the research.

☐ No, I do not consent.

**Participant's full name:**
**Participant's signature:**
**Date:**

**Full name of researcher present:**
**Researcher's signature:**
**Date:**

The researcher declares that the participant has received extensive information about the research.

*You have the right to a copy of this consent form.*

## Appendix B. Information sheet

# Information sheet – session 1

**INSTRUCTIONS**

In this experiment, you will work with CTV mesorectum auto-delineations of 10 patients. You are asked to review these delineations, edit if needed and fill out the questionnaire. Please perform the tasks as you were in an **online adaptive workflow** and follow **your patients order**.

We kindly request to record your screen all the time while performing any tasks related to this experiment. We will use it to extract the time taken to perform the delineation task.

Please ensure that you complete all tasks independently, without any influence from other participants. Lastly, **it is crucial that you follow the instructions precisely for each case to avoid any potential differences in visualization.**

In the following, you will find three main sections.

1. **HOW TO ACCESS THE SHARED FOLDER**: instructions on how to access and use the shared folder, along with a description of its contents.
2. **HOW TO RECORD YOUR SCREEN:** instructions on how to start, end and save a video recording of the screen.
3. **TASKS:** step-by-step instructions for completing the required tasks for each case in this experiment.

## 1. HOW TO ACCESS THE SHARED FOLDER

1. Open *Teams.*

2. Go to *OneDrive* (1). If you do not find it, click on the three dots and search for it (2). Then open the shared file *Project uncertainty*.

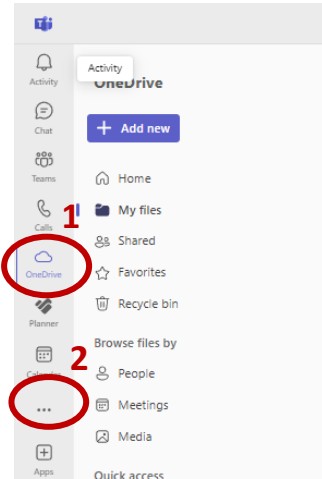

3. Select the folder *Phase 1* and then open the folder with your name. There you will find the questionnaire, a copy of this information sheet, and several folders. You will use the folder *Video recordings* to upload the recordings you create during the experiments and the folder *Questionnaire completed* to upload the questionnaire once you are done with this phase. You won't be using the folder *RTstruct&sessions*. I will take care of transferring your edited structures from Mirada to OneDrive.

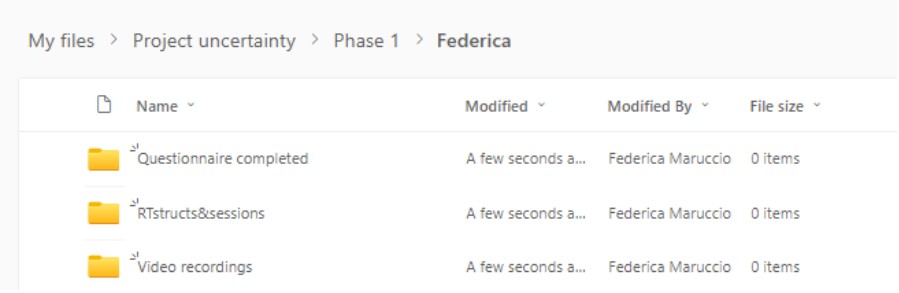

## 2. HOW TO RECORD YOUR SCREEN

### Start a video recording

1. Open the screen recording website (https://www.microsoft365.com/launch/stream?auth=2) and press *Screen recording.*

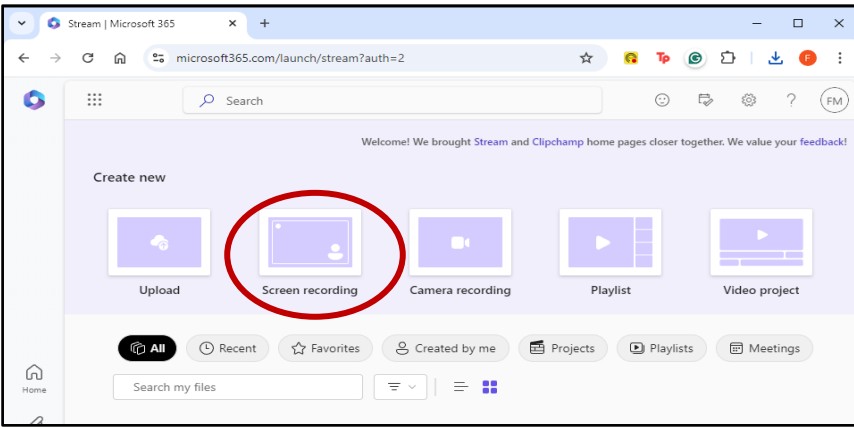

2. Close the camera window by clicking on the *x* (1) and disable the microphone (2). Then click on *Start screen recording* (3).

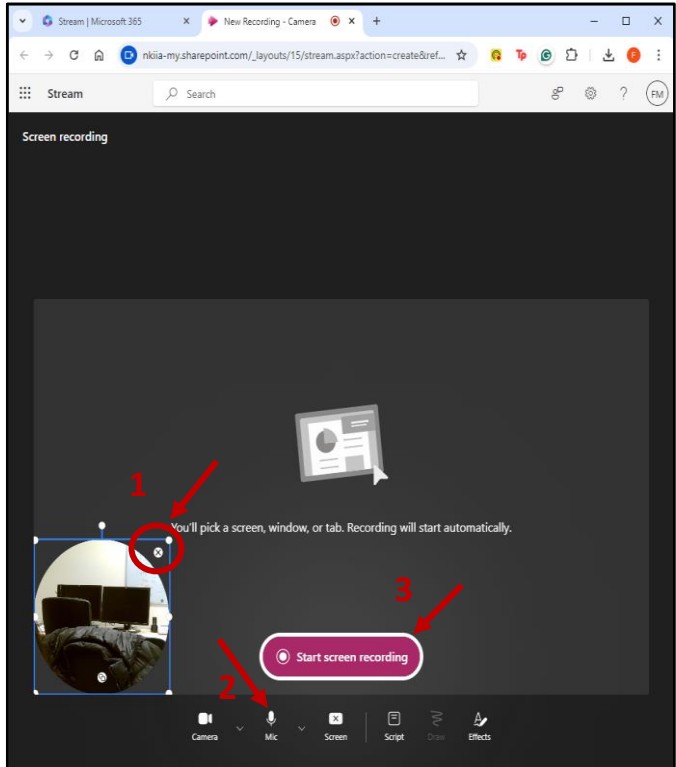

3. Choose entire screen (1), then select the screen where Mirada is open and will be used (2). (If still not open, please launch the program). Click *Share* (3).

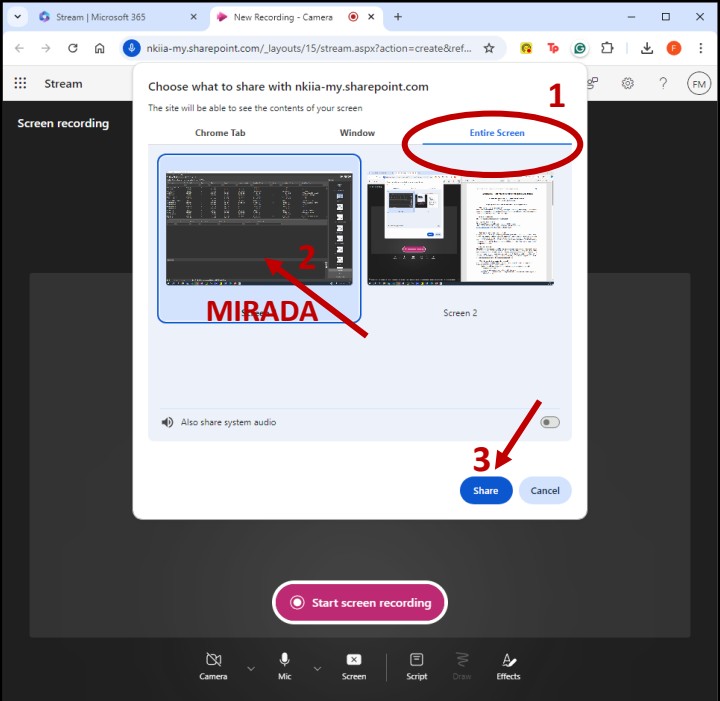

4. All the tasks you perform on this screen will be recorded. **Please, pay attention to not moving the Mirada window to another screen while recording. It is important that the recording captures the delineation task.** Now you are ready to start editing!

## End a video recording

5. When you are done with the tasks for that patient, go to the screen recording tab and click *Review to* check the video recording. If everything is okay, click *Finish*.

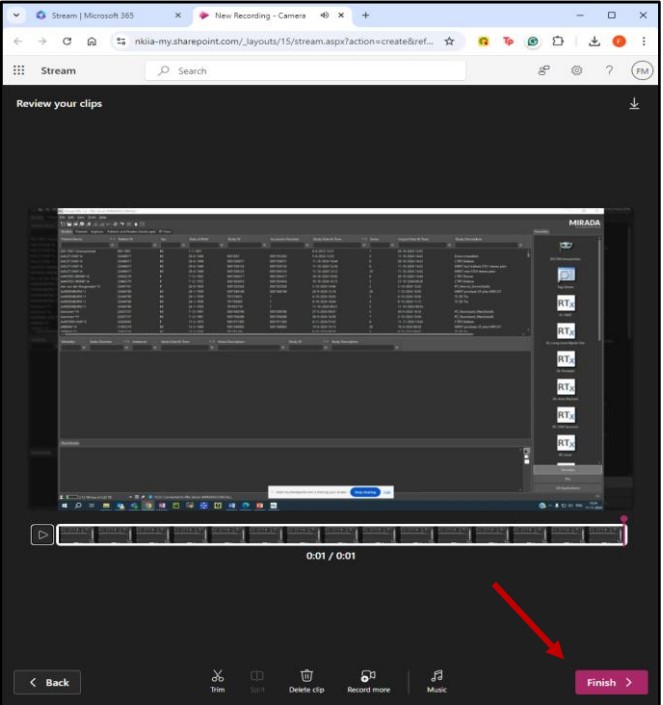

6. Download the video recording by clicking on the three dots above the video (1) and then selecting '*Download*' (2). The download will appear in your Download folder.
   Please rename the file as '*Video_phase1_<your name_<patient_number>*' and copy it to the shared folder.

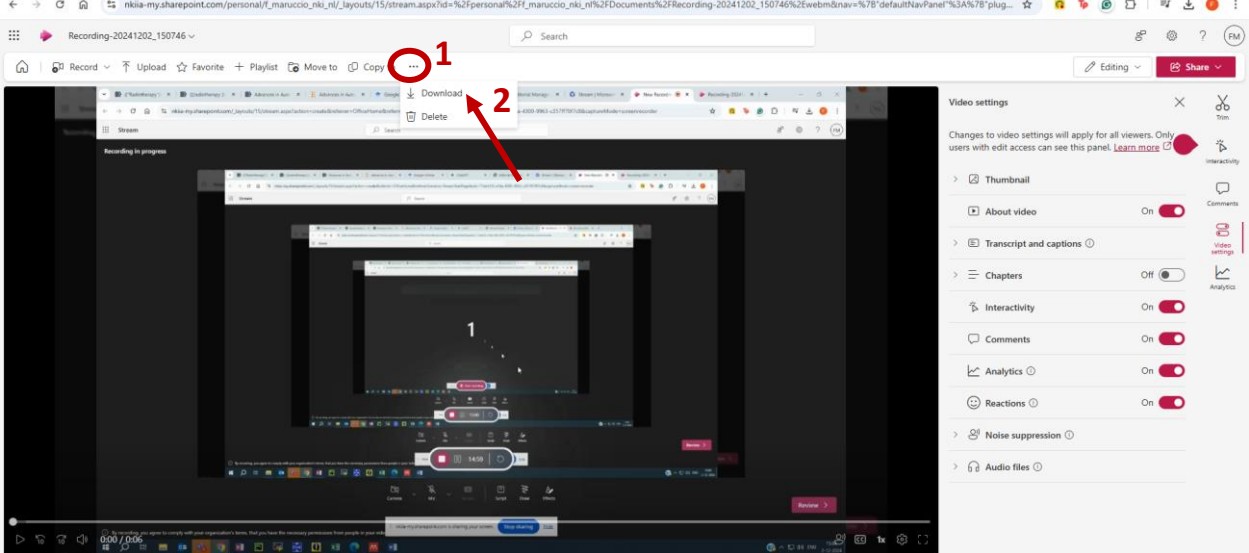

**Note:** We kindly suggest testing the recording system the first time you use it by going through all the previous steps before starting the actual experiment.

## 3. TASKS

1. Download the Word file *Questionnaire1* from the shared folder and open it.

2. Open Mirada and choose the database *MIRADADEV*.

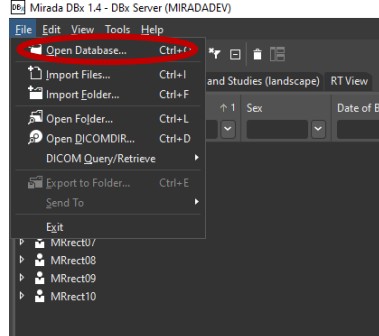 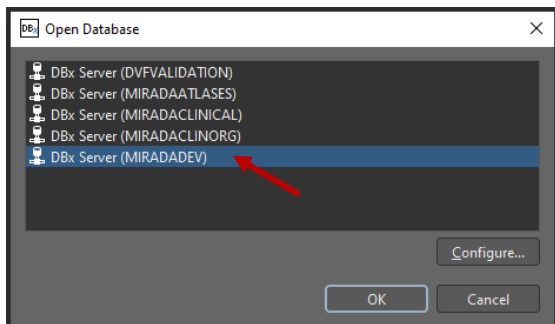

3. Search for the data with the tag *Federica* using the *Tags* searching bar. You should see a list of 10 patients called *MRrect*.

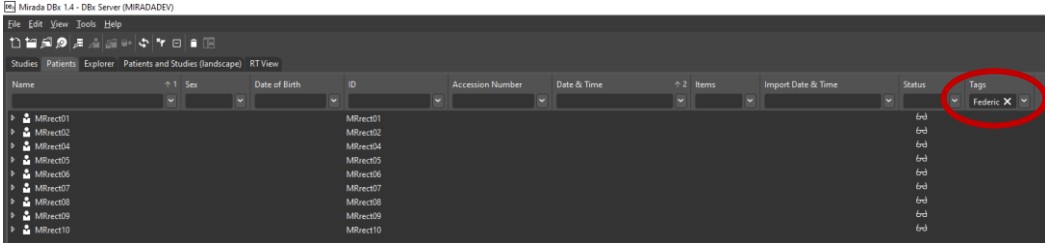

4. Start the screen recording. (Refer to section *How to record*, steps 1-4).

5.
   - In case of a patient with only DL contour available:
     Open the patient by selecting both the MR image and the RTSTRUCT and choosing the ***Gyn brachy Gammaknife*** **viewer**, as shown in the image below. Ignore the sessions saved by the other participants.

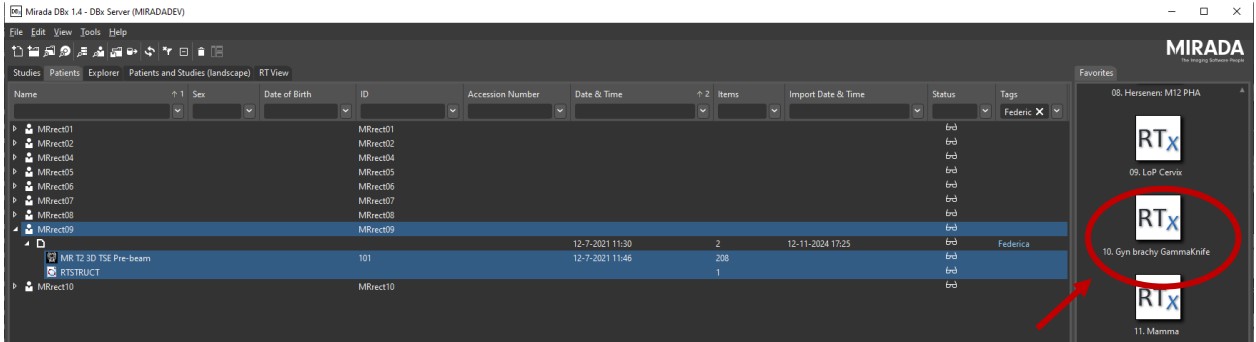

- In case of a patient with both DL contour and uncertainty map available:

Open the patient by selecting the MR image, the RTSTRUCT and the uncertainty map and choosing the **Gyn brachy Gammaknife** viewer, as shown in the image below. Ignore the sessions saved by the other participants.

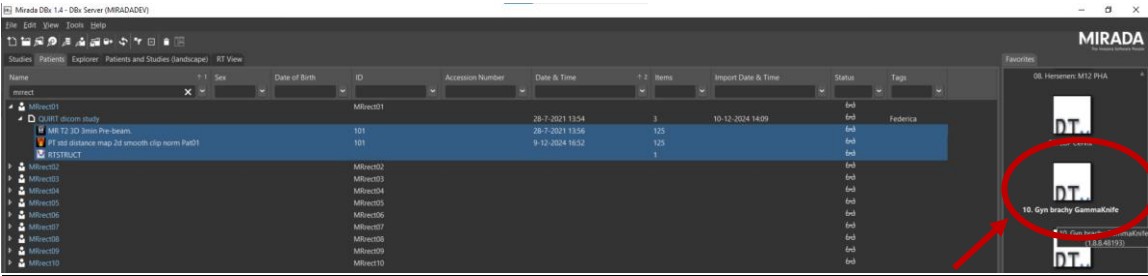

6.

- In case of a patient with only DL contour available:

Drag and drop the MR image to the *MR planning* box.

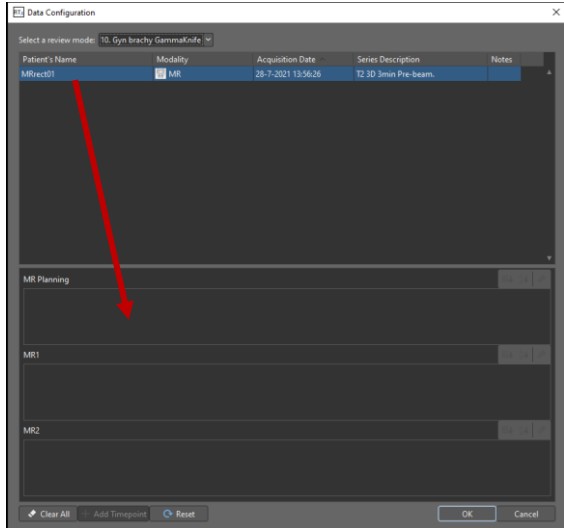

- In case of a patient with both DL contour and uncertainty map available:

Drag and drop the MR image to the *MR planning* box and the uncertainty map to *MR1* box.

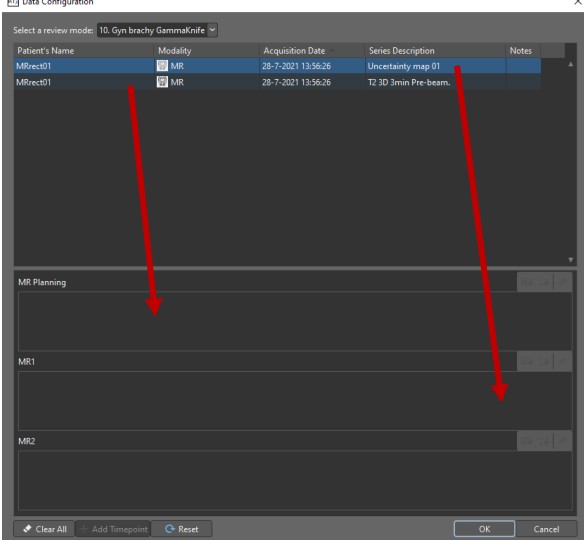

7. Only in case of a patient with both DL contour and uncertainty map available:
   Open the ***visualization*** tool by clicking on *Image* in the taskbar (1). Then:
   - select ***MR1*** (2)
   - select ***Min-75% Range*** for the intensity values (3)
   - choose the ***green-magenta*** colorbar (4)
   - change the **transparency to 30%** (5)
   - Click *OK*.

     **Please follow these steps precisely for each patient.**

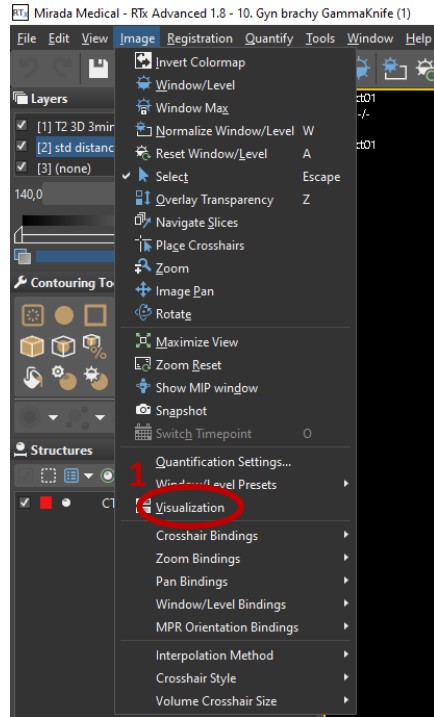

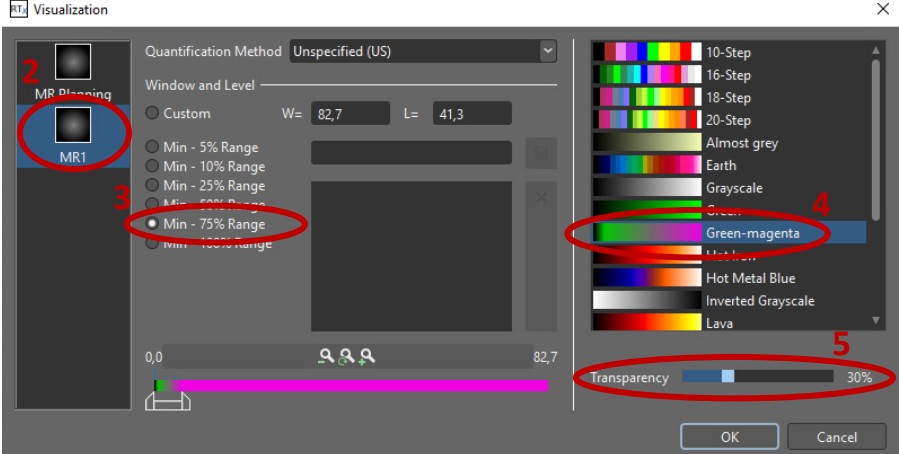

(In case Mirada opens in registration mode instead of the standard viewer, press 'Back to layout' and then perform step 7).

8.  Select the CTV in the *Structures* box (1) and edit if needed by using the contouring tools. You don't need to copy the structure, you can directly edit the CTV as it is. You can select/deselect the visualization of the automatic delineation (2) and of the uncertainty map (3), if present. You can use the *space bar* as well to select/deselect the uncertainty map visualization.

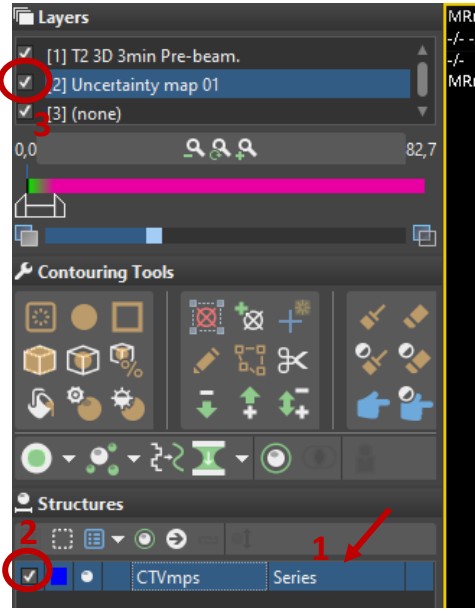

9.  Once you are done with this patient, save the whole session (1) typing '*Phase1_<your name>*' in the series description (2) and selecting *Mirada DBx* as a destination (3).

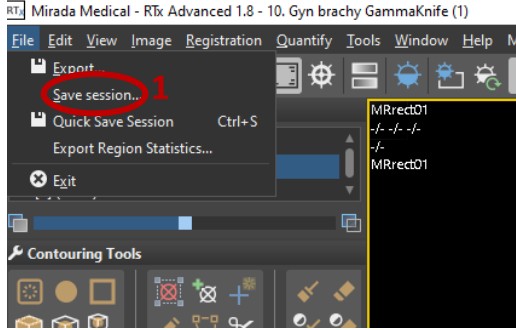 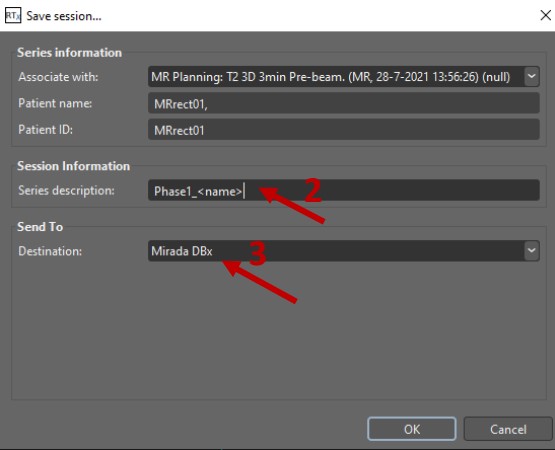

10. In the Word file *Questionnaire1,* answer the two questions (Q1 – Q2) in Table 1 related to this patient, for example, row P1.

| Patients | Q1. How do you rate the unedited model's prediction? (1 - 4) | Q2. How confident are you in your final decision? (1 - 5) |
|---|---|---|
| | 1. Not acceptable: re-delineation needed
2. Acceptable: usable but many slices (>5) corrected
3. Good: limited number of slices (2-5) corrected
4. Excellent: almost no modification | 1. Totally unconfident
2. Somewhat unconfident
3. Indecisive
4. Somewhat confident
5. Fully confident |
| P1 | | |

11. Stop the video recording and save it (Refer to section *How to record,* steps 5-6).

12. Repeat steps 4-11 for all the 10 patients following your assigned patients order.

**In case of any questions, please reach out to *Federica Maruccio* via Teams or email (f.maruccio@nki.nl).**

## Appendix C. Patient order

| Session | Participant | 1st | 2nd | 3rd | 4th | 5th | 6th | 7th | 8th | 9th | 10th |
|---------|-------------|-----|-----|-----|-----|-----|-----|-----|-----|-----|------|
| Session 1 | A | P1 | P5 | P4 | P2 | P3 | P7 | P6 | P8 | P9 | P10 |
| | B | P3 | P2 | P1 | P5 | P4 | P6 | P7 | P10 | P8 | P9 |
| | C | P4 | P2 | P1 | P3 | P5 | P9 | P6 | P10 | P8 | P7 |
| | D | P2 | P5 | P3 | P1 | P4 | P6 | P10 | P7 | P8 | P9 |
| | E | P3 | P1 | P2 | P4 | P5 | P10 | P6 | P9 | P7 | P8 |
| | F | P5 | P2 | P3 | P4 | P1 | P8 | P9 | P7 | P6 | P10 |
| Session 2 | A | P6 | P10 | P7 | P8 | P9 | P3 | P1 | P2 | P4 | P5 |
| | B | P8 | P10 | P6 | P7 | P9 | P5 | P1 | P4 | P3 | P2 |
| | C | P7 | P10 | P9 | P8 | P6 | P2 | P1 | P4 | P5 | P3 |
| | D | P9 | P7 | P8 | P6 | P10 | P1 | P4 | P3 | P2 | P5 |
| | E | P7 | P9 | P10 | P6 | P8 | P2 | P5 | P3 | P1 | P4 |
| | F | P10 | P9 | P6 | P8 | P7 | P5 | P2 | P3 | P4 | P1 |

Table 2: Patient presentation order for each participant in session 1 and session 2.

# Appendix D.  Questionnaires

## Questionnaire session 1

Let's assume we are in an **online adaptive workflow.**

Look at the structure, edit if needed, and answer the questions in the table for each of the 10 cases (P1 – P10) by filling out the table. **Please follow your assigned patients order.**

In five cases you will have only the DL contour, and in the remaining 5 you will be provided with the DL contour and the uncertainty map.

| | DL contour only |
|---|---|
| | DL contour + uncertainty |

| Patients | Q1. How do you rate the unedited model's prediction?  (1 - 4)

1.  Not acceptable: re-delineation needed
2.  Acceptable: usable but many slices (>5) corrected
3.  Good: limited number of slices (2-5) corrected
4.  Excellent: almost no modification

(Edit counts if >1mm) | Q2. How confident are you in your final decision?  (1 - 5)

1.  Totally unconfident
2.  Somewhat unconfident
3.  Indecisive
4.  Somewhat confident
5.  Fully confident |
|---|---|---|
| P1 | | |
| P2 | | |
| P3 | | |
| P4 | | |
| P5 | | |
| P6 | | |
| P7 | | |
| P8 | | |
| P9 | | |
| P10 | | |

## Questionnaire session 2

Let's assume we are in an **online adaptive workflow.**

Look at the structure, edit if needed, and answer the questions in the table for each of the 10 cases (P1 – P10). **Please follow your assigned patient's order.**

In five cases you will have the DL contour only, in the remaining 5 you will be provided with the DL contour and the uncertainty map.

When you have edited the contours of all cases, answer the final questions at the end of this document.

| | |
|---|---|
| | DL contour + uncertainty |
| | DL contour only |

| Patients | Q1. How do you rate the unedited model's prediction? (1 - 4)

1. Not acceptable: re-delineation needed
2. Acceptable: usable but many slices (>5) corrected
3. Good: limited number of slices (2-5) corrected
4. Excellent: almost no modification

(Edit counts if >1mm) | Q2. How confident are you in your final decision? (1 - 5)

1. Totally unconfident
2. Somewhat unconfident
3. Indecisive
4. Somewhat confident
5. Fully confident |
|---|---|---|
| P1 | | |
| P2 | | |
| P3 | | |
| P4 | | |
| P5 | | |
| P6 | | |
| P7 | | |
| P8 | | |
| P9 | | |
| P10 | | |

**FINAL QUESTIONS**

1. **DL CONTOUR ONLY**

Indicate to what extent you agree with the following statements, concerning the provision of DL contours only.

Keep in mind: **provided information = DL contour**

|  | Strongly Disagree |  |  |  |  |  | Strongly Agree |
|---|---|---|---|---|---|---|---|
| I find the provided information complex. | 1 | 2 | 3 | 4 | 5 | 6 | 7 |
| I feel like the provided information helps me. | 1 | 2 | 3 | 4 | 5 | 6 | 7 |
| I prefer seeing the provided information over delineating from scratch. | 1 | 2 | 3 | 4 | 5 | 6 | 7 |
| The provided information confuses me. | 1 | 2 | 3 | 4 | 5 | 6 | 7 |
| I think using the provided information as a basis for my delineations could save time. | 1 | 2 | 3 | 4 | 5 | 6 | 7 |
| I think it is feasible to use the provided information in the clinic. | 1 | 2 | 3 | 4 | 5 | 6 | 7 |
| I prefer manual delineation over using such information. | 1 | 2 | 3 | 4 | 5 | 6 | 7 |
| Using the provided information would make me more confident in my delineations. | 1 | 2 | 3 | 4 | 5 | 6 | 7 |
| I was influenced in my decision-making by the information provided. | 1 | 2 | 3 | 4 | 5 | 6 | 7 |

## 2. DL CONTOUR + UNCERTAINTY MAP

Indicate to what extent you agree with the following statements concerning the provision of **DL contours + uncertainty maps.**

Keep in mind: **provided information = DL contour + uncertainty map**

| | Strongly Disagree | | | | | | Strongly Agree |
|---|---|---|---|---|---|---|---|
| I find the provided information complex. | 1 | 2 | 3 | 4 | 5 | 6 | 7 |
| I feel like the provided information helps me. | 1 | 2 | 3 | 4 | 5 | 6 | 7 |
| I prefer seeing the provided information over seeing only the predicted contour. | 1 | 2 | 3 | 4 | 5 | 6 | 7 |
| The provided information confuses me. | 1 | 2 | 3 | 4 | 5 | 6 | 7 |
| I think using the provided information as a basis for my delineations could save time. | 1 | 2 | 3 | 4 | 5 | 6 | 7 |
| I think it is feasible to use the provided information in the clinic. | 1 | 2 | 3 | 4 | 5 | 6 | 7 |
| I prefer auto-segmentation only over using such information. | 1 | 2 | 3 | 4 | 5 | 6 | 7 |
| Using the provided information would make me more confident in my delineations. | 1 | 2 | 3 | 4 | 5 | 6 | 7 |
| I was influenced in my decision-making by the information provided. | 1 | 2 | 3 | 4 | 5 | 6 | 7 |

# Appendix E. Additional results

## E.1. Editing time

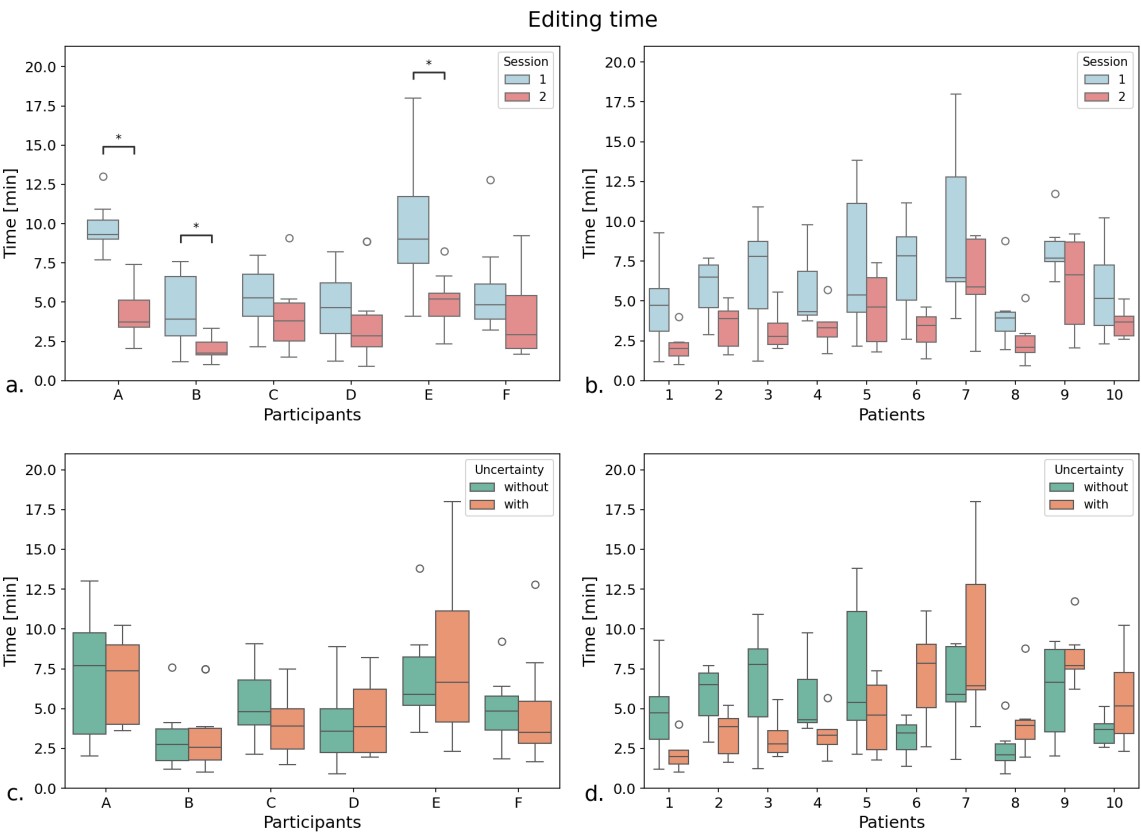

Figure 7: Participant- and patient-level editing time comparison between conditions (c-d) and sessions (a-b).

## E.2. Geometric metrics

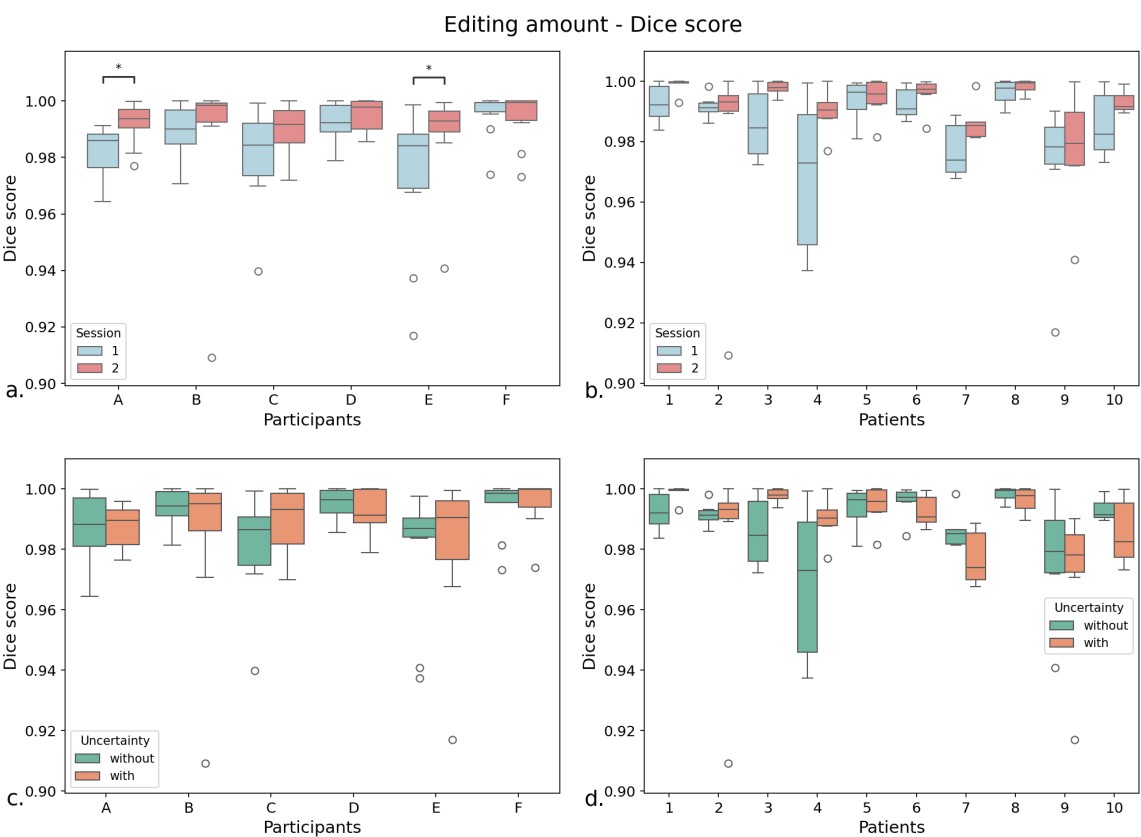

Figure 8: Participant- and patient-level Dice values comparison between conditions (c-d) and sessions (a-b).

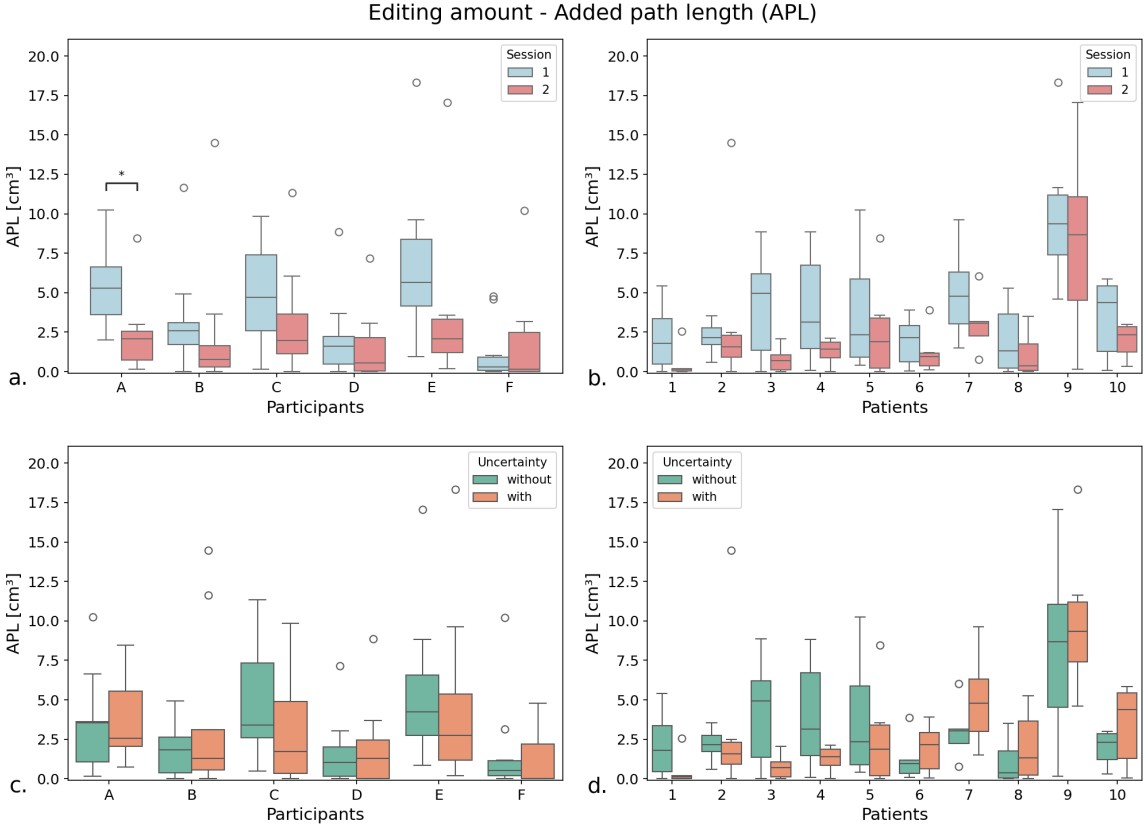

Figure 9: Participant- and patient-level APL values comparison between conditions (c-d) and sessions (a-b).

## E.3. Questionnaires

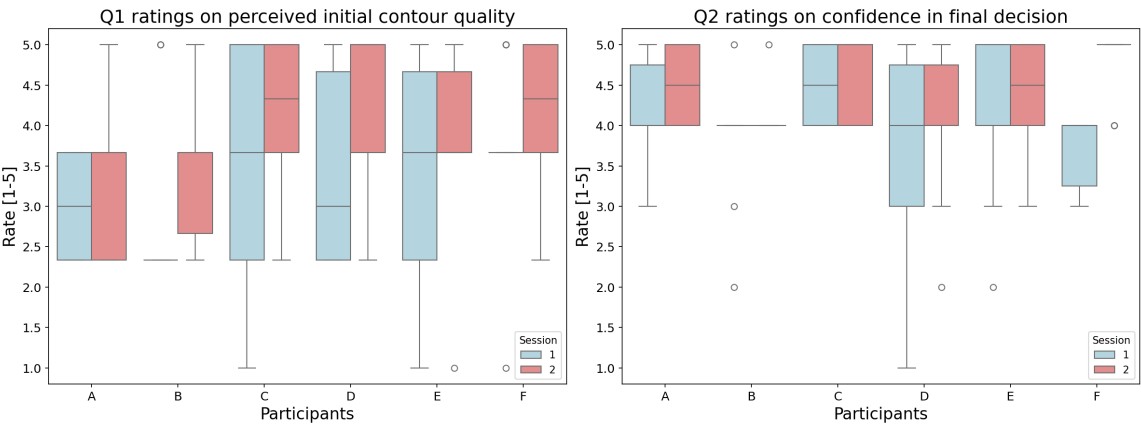

Figure 10: Participant-level Q1 and Q2 responses across sessions.

Table 3: Responses to question Q1 (*How do you rate the unedited model's prediction?*[1-4]) for all participants (A-F) and patients (P1-P10), in sessions 1 and 2.

| Participant | Session | P1 | P2 | P3 | P4 | P5 | P6 | P7 | P8 | P9 | P10 |
|---|---|---|---|---|---|---|---|---|---|---|---|
| A | 1 | 3 | 3 | 2 | 2 | 2 | 3 | 2 | 2 | 3 | 3 |
|   | 2 | 3 | 3 | 3 | 2 | 2 | 3 | 2 | 3 | 4 | 2 |
| B | 1 | 4 | 2 | 2 | 2 | 2 | 2 | 2 | 2 | 2 | 4 |
|   | 2 | 4 | 2 | 2 | 3 | 3 | 3 | 3 | 4 | 2 | 3 |
| C | 1 | 4 | 3 | 2 | 2 | 4 | 4 | 1 | 3 | 2 | 4 |
|   | 2 | 4 | 3 | 4 | 3 | 4 | 3 | 3 | 4 | 2 | 4 |
| D | 1 | 4 | 2 | 4 | 3 | 3 | 2 | 2 | 4 | 2 | 2 |
|   | 2 | 4 | 3 | 4 | 4 | 4 | 4 | 2 | 4 | 2 | 3 |
| E | 1 | 4 | 3 | 3 | 1 | 3 | 4 | 2 | 4 | 1 | 2 |
|   | 2 | 4 | 3 | 4 | 3 | 3 | 4 | 3 | 3 | 1 | 3 |
| F | 1 | 3 | 3 | 3 | 3 | 3 | 3 | 1 | 4 | 3 | 4 |
|   | 2 | 4 | 4 | 4 | 4 | 3 | 3 | 2 | 4 | 2 | 3 |

Table 4: Responses to question Q2 (*How confident are you in your final decision?* [1-5]) for all participants (A-F) and patients (P1-P10), in sessions 1 and 2.

| Participant | Session | P1 | P2 | P3 | P4 | P5 | P6 | P7 | P8 | P9 | P10 |
|---|---|---|---|---|---|---|---|---|---|---|---|
| A | 1 | 4 | 5 | 5 | 4 | 4 | 5 | 3 | 4 | 3 | 4 |
|   | 2 | 5 | 5 | 5 | 4 | 4 | 5 | 4 | 4 | 4 | 5 |
| B | 1 | 5 | 4 | 4 | 4 | 4 | 4 | 4 | 4 | 2 | 3 |
|   | 2 | 5 | 4 | 4 | 4 | 4 | 4 | 4 | 4 | 4 | 4 |
| C | 1 | 5 | 4 | 4 | 4 | 5 | 5 | 4 | 5 | 4 | 5 |
|   | 2 | 5 | 5 | 4 | 5 | 5 | 4 | 4 | 5 | 4 | 5 |
| D | 1 | 4 | 4 | 5 | 5 | 3 | 3 | 3 | 5 | 4 | 1 |
|   | 2 | 5 | 4 | 5 | 4 | 4 | 4 | 4 | 5 | 3 | 2 |
| E | 1 | 5 | 5 | 4 | 3 | 2 | 4 | 4 | 5 | 5 | 4 |
|   | 2 | 5 | 4 | 5 | 4 | 3 | 5 | 4 | 5 | 4 | 5 |
| F | 1 | 3 | 4 | 4 | 3 | 4 | 4 | 4 | 3 | 4 | 4 |
|   | 2 | 5 | 5 | 4 | 5 | 4 | 5 | 5 | 5 | 5 | 5 |

Table 5: Responses to the final questions for conditions *Contour only* and *Contour + uncertainty* across all participants (A-F).

| Condition | Subquestion | A | B | C | D | E | F |
|---|---|---|---|---|---|---|---|
| Contour only | complex | 1 | 1 | 2 | 1 | 1 | 1 |
| | helpful | 6 | 2 | 6 | 7 | 7 | 6 |
| | preferable | 7 | 5 | 6 | 7 | 7 | 7 |
| | confusing | 2 | 2 | 2 | 1 | 1 | 1 |
| | time-saving | 7 | 5 | 6 | 7 | 7 | 7 |
| | feasible | 7 | 4 | 6 | 7 | 7 | 7 |
| | preferable reversed | 1 | 2 | 2 | 1 | 1 | 1 |
| | increasing confidence | 2 | 2 | 3 | 4 | 5 | 2 |
| | influencing DM | 2 | 3 | 3 | 7 | 6 | 2 |
| Contour + uncertainty | complex | 2 | 1 | 2 | 3 | 1 | 1 |
| | helpful | 4 | 2 | 6 | 5 | 2 | 4 |
| | preferable | 4 | 5 | 4 | 2 | 2 | 3 |
| | confusing | 2 | 2 | 2 | 7 | 1 | 1 |
| | time-saving | 4 | 5 | 5 | 2 | 2 | 2 |
| | feasible | 4 | 4 | 5 | 4 | 4 | 5 |
| | preferable reversed | 3 | 2 | 4 | 7 | 6 | 6 |
| | increasing confidence | 4 | 1 | 2 | 4 | 2 | 6 |
| | influencing DM | 3 | 1 | 3 | 7 | 2 | 4 |

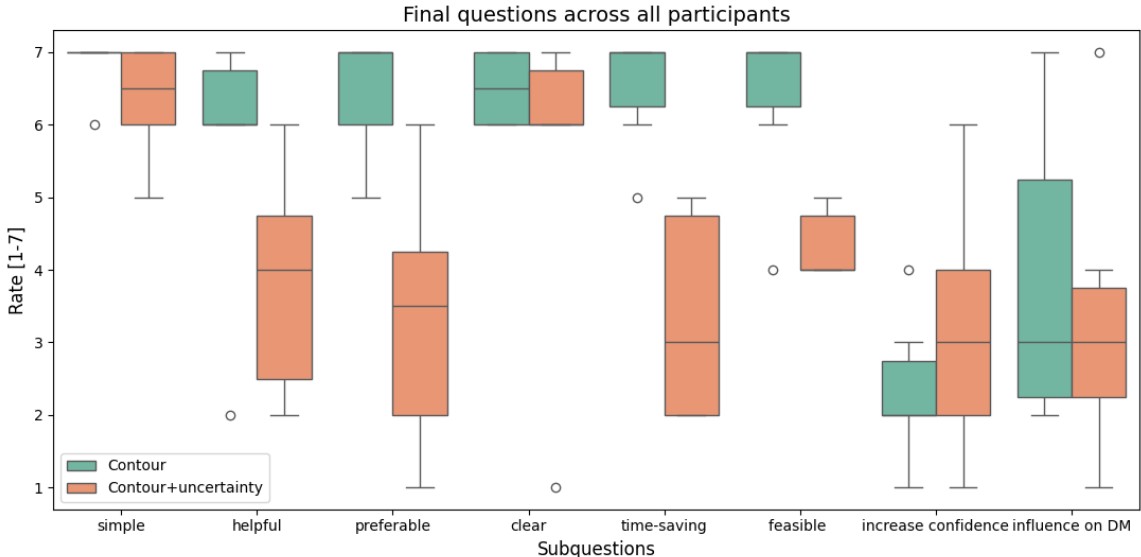

Figure 11: User-experience questionnaire responses across conditions. All the negatively phrased items were reverse-scored.

### E.4. Interviews

E.4.1. Interview summary participant A

The participant reported that the provided contours were already of good quality, although considerable adjustments were still performed during the first study session. This behaviour was attributed to working habits: in treatment planning, the participant delineates manually from scratch and therefore tended to refine the contours extensively at the beginning of the study. The participant explained that it took time during the study to get used to editing an automatic contour rather than delineating it from scratch. Over the course of the study, she recognised that very small adjustments have minimal dosimetric impact as long as the tumour is adequately covered. Consequently, a learning curve was recognised, which contributed to faster editing during the second session. The participant emphasised that the novelty of using auto-segmentation applied to both the contours and the uncertainty map, whereas the RTT participants were already more familiar with such a workflow. Moreover, the participant reported using Mirada for delineating during treatment planning, therefore no learning effect applied there.

Regarding the uncertainty map, the magenta areas were perceived as influencing the editing process sometimes, particularly when confused about a region. The green areas were perceived as less influential because she tended to edit these regions regardless, partly due to own confidence and decisiveness. The uncertainty map was used initially to obtain an overall impression of the segmentation and then sometimes consulted during the editing. The participant indicated that the additional information is helpful, particularly for clinicians used to full manual delineation, as it may facilitate trust-building in the auto-segmentation model.

The participant found the magenta areas clear, but reported that the lack of a grey for the intermediate uncertainty range hindered visual clarity. The participant emphasised that time pressure in an online adaptive workflow contrasts with offline planning, where delineations can be more meticulous. The participant highlighted that planning contours produced by clinicians must be highly accurate because they serve as ground truth for the RTTs in subsequent fractions. For RTTs, pragmatic decisions may be made based on available time. In the study setting, the participant was required to delineate the CTV without knowledge of the gross tumour volume (GTV) location, which was described as unusual in clinical practice. As a result, some adjustments in session 1 were guided by hypothetical tumour location. This effect diminished in session 2 as a more pragmatic, online-workflow mindset was adopted.

Increased clinical workload during June, combined with the upcoming holiday period, was acknowledged as a potential influencing factor in session 2. The participant did not recall any influence related to remembering the patients' anatomy from earlier sessions. Regarding the influence of the uncertainty map on the amount of editing, the participant stated that the map attracted attention to specific regions, although the final decision always relied on their own judgment. The participant underscored that trust in the map is essential for time savings; consistent exposure and confirmation that the map is reliable could enable clinicians to avoid editing green areas and focus primarily on magenta regions. The participant believed that greater benefit might be obtained in more complex cases than those included in the study.

The participant also expressed the view that uncertainty information may be more beneficial in online workflows due to stronger time constraints. The participant noted that RTTs may hesitate to rely on uncertain cues if these signals do not clearly originate from the physician's delineation, which could explain their preference for offline rather than online usage. Finally, the participant suggested that the visualisation could be improved by enhancing the distinction between grey and the green/magenta regions, and recommended providing easier visualisation options, such as displaying only high-uncertainty areas based on a user-defined threshold. Switching the map on and off was experienced as inefficient during the first session, but was less problematic in the second.

### E.4.2. Interview summary participant B

The participant reported that the overall quality of the contours was already high, requiring only minor adjustments during the task. The uncertainty map was used selectively rather than continuously: it was typically activated only when the participant encountered doubts regarding specific regions. In such cases, if the region displayed high uncertainty, the participant proceeded with editing. Otherwise, the map remained off for most of the editing process. The participant indicated that the map was useful for confirming uncertainties in localised areas, but was not considered necessary for the entire structure.

The participant had prior experience with the delineation software Mirada due to involvement in post-processing tasks, which contributed to confidence in using it. From the participant's perspective, having a DL contour was generally sufficient, and the uncertainty map did not add substantial value. This was partly because, in routine clinical practice, RTTs typically work with a reference planning contour provided by a RadOnc, which already serves as a reliable guide. As a result, the participant believed the uncertainty map might offer more benefit to RadOncs, who delineate from scratch during treatment planning, than to RTTs. The participant further suggested that the map could be more relevant in complex or less accurate cases, but was less beneficial for the relatively straightforward cases used in this study.

There was no perceived difference in confidence between delineating with or without the uncertainty map. The participant reported using the map minimally, often only at the beginning of a case before turning it off. Consequently, the participant did not perceive a strong influence of the map on the amount of editing performed. Furthermore, a shift in mindset between the two experimental sessions was reported. Initially (session 1), the participant felt an implicit expectation to make modifications to the contour. In session 2, the participant adopted a more pragmatic approach and accepted that the auto-segmentation was generally adequate. This shift was also attributed to concurrent clinical responsibilities during that period: the participant was more involved in long treatment series where time efficiency is essential, and small geometric deviations are often deprioritised. This difference in mentality, rather than workload, may have contributed to behavioural changes across sessions. The participant did not perceive that the second session was easier due to familiarity with the cases.

In clinical practice, the participant described a standard workflow wherein the RTT seeks a second opinion from a RadOnc or colleague when uncertain. When possible, delineations are checked or performed by two RTTs. The participant also noted a personal tendency that, once editing begins, there is an inclination to continue adjusting more extensively.

Moreover, no issues were reported regarding the visualisation of the uncertainty map, and no improvements were suggested. Finally, the participant acknowledged occasionally forgetting to turn on the map.

### E.4.3. Interview summary participant C

The participant reported frequently switching the uncertainty map on and off to visualise the underlying anatomical structures, a workflow similar to how the participant typically handles DL contours alone. During clinical work, the participant actively recalls comments from RadOncs and attempts to incorporate those considerations, while during this study, such information was not available. However, for the cases presented in the study, only minor adjustments were generally required, in their opinion.

The participant described a habit of editing by deleting slices, interpolating, and then reviewing each slice to verify the interpolation results. The uncertainty map was used primarily in situations of doubt rather than for every slice, as overlapping contours and colour overlays could obscure the precise border. When using the map, the participant sometimes revisited slices to avoid being overly influenced by the visualisation.

According to the participant, delineation behaviour can vary depending on one's mood. On some days, the task may be approached more pragmatically, while on others with greater attention to detail. The participant noted that starting in an "editing mode" often leads to extensive editing, even when slices are adequate, sometimes motivated by a desire for a smoother structure. The participant also felt slower when using the map due to the additional interaction steps involved.

The magenta regions tended to attract the participant's attention and motivated a closer inspection, although the final decisions were based primarily on personal judgment. Green areas were interpreted as normal and were therefore largely ignored since they occurred more frequently. The participant did not perceive any difference in confidence between using and not using the map, emphasising that confidence is strongly tied to the anatomical complexity of each patient. The participant was also unsure whether the map resulted in any time savings overall.

The participant acknowledged a tendency to spend more time on the first cases of a session or study, becoming more pragmatic with subsequent cases once a clearer understanding of clinically relevant adjustments is developed. A similar pattern is followed in clinical practice, where the first fraction of a patient usually takes longer, and subsequent fractions become more efficient due to prior feedback. Furthermore, the participant reported remembering the anatomy of all patients due to strong visual memory, but not the editing performed. In the second session, this familiarity contributed to a clearer understanding of what needed to be done, reflecting a learning curve. Particular attention was paid to anatomical regions where recurrence risk is higher, especially near the bones, based on training received from the doctors.

A key difference from clinical practice was the absence of the physician's reference contour during the experiment. This lack of reference introduced additional uncertainty, as decisions normally guided by the physician's input had to be made independently. The participant stated that occasional oversights between sessions are normal and can also occur in routine clinical workflows. Moreover, the absence of GTV contours made caudal extent decisions more challenging. The participant also reported sometimes forgetting to use the uncertainty

map and only activating it upon remembering its importance in the study; therefore, differences observed between sessions could not be attributed to the map. The participant also noted that when in "editing mode", it can be faster to correct minor issues immediately rather than leave them to avoid accumulating multiple wrong slices.

Upon reviewing the video recordings, the participant perceived that using the map might have been distracting, shifting attention from the primary editing task to the act of checking the map, especially due to the frequent switching required. Moreover, it was reported that in general, when the provided contour is good, there is more time to address fine-grained adjustments, but for more complex anatomy, the priority would shift to clinically relevant corrections, while small details might be deprioritised.

Furthermore, the second study session occurred during a period of higher workload in the clinic compared to the first session, making time pressure more acute. This was believed to determine a more pragmatic editing style, consistent with clinical practice, where busy periods prompt a focus on essential corrections only. Moreover, the participant declared not to have discussed cases with the other participants between sessions.

The participant suggested that the map may have slightly influenced the amount of editing, particularly that green areas might have encouraged faster navigation, although this effect was modest. Ultimately, confidence in the map was considered essential: if reliability were assured, the map could meaningfully support efficiency. However, if the map deviated from the physician's planning contours, it could prompt additional discussion, as clinical workflows prioritise adherence to physician-defined structures. The participant suggested that such maps could be especially useful for the RadOncs during planning to support more consistent delineations, which would benefit RTTs who are trained to follow established rules and protocols.

For visualisation improvements, the participant recommended displaying two views next to each other, one with the map and one without, so that the map could be consulted only when needed without constant toggling. The participant noted that integration in Monaco, the other delineation software used in the clinic, may be challenging, given the lack of keyboard shortcuts that already pose practical limitations. The participant also highlighted that the study cases were relatively simple with good-quality initial contours. In more complex cases, the map could be more beneficial, provided that its reliability is well established. However, the participant expressed doubts about the map's ability to support extremely challenging scenarios.

### E.4.4. Interview summary participant D

The participant reported difficulties with the transparency of the uncertainty map and found the overlay obstructive, particularly because the visualisation covered the delineation boundary. As a result, the participant frequently switched the map off, at which point the map was perceived as no longer useful since the underlying border was already clear. The participant also noted that the green and magenta colours used in the map conflicted with their meaning in XVI, software used in the clinic for image registration, which caused confusion.

Differences between the two study sessions were attributed primarily to variations in mindset and daily conditions. The participant described this as comparable to clinical practice, where time pressure and patient waiting times can influence the approach to delineation.

The clinical workload was higher in June than in March, and the participant expressed a desire to complete the delineations more quickly in the second session.

The participant was not always able to assess the influence of the maps and often responded "I don't know", but later stated that the maps did not affect the amount of editing, largely because they were not used often. In cases where confidence in the contour was low and the map also indicated high uncertainty, the participant often chose to leave the contour unchanged.

The participant was aware of performing fewer edits in the second session. After session 1, some RTTs discussed the cases with each other, and the participant acknowledged that this discussion influenced perceptions and decisions in session 2. Moreover, the participant remembered the anatomy of approximately three cases.

In clinical workflows, RTTs routinely compare their contours with the physician's planning CT. The absence of such a reference contributed to confusion during session 1. By session 2, the participant had adjusted expectations and was more confident despite the lack of comparison. The participant suggested that, because clinical practice already includes the reference from the doctor, the uncertainty map would not be necessary for experienced RTTs. However, it might be beneficial for less-experienced RTTs or for doctors at the planning stage, who do not have a direct comparison.

The absence of additional clinical information, such as patient gender, was also reported as a source of confusion in session 1, though this became less problematic in session 2. No experience with Mirada also introduced a learning component to the first session. Moreover, the participant described the overall contour quality as very good, which reduced the need for the uncertainty map. When errors were large and obvious, the participant felt the map was unnecessary; when differences were small, the map did not add value, although usefulness was considered dependent on the clinician's level of experience.

Session 1 required adaptation to the study workflow, including adjusting to the software interface, the recording setup, and the questionnaire. By session 2, these elements were familiar, and the participant had accepted the absence of a clinical reference. The participant expressed a preference for using only the DL contour and felt that the current implementation of the map provided limited added value. Nevertheless, the participant considered the voxel-level visualisation more useful than receiving only a numerical uncertainty score. Finally, when reviewing the results, the participant confirmed the presence of memory bias, different mindsets across sessions, differences in workload, and an overall perception that the second session was easier.

### E.4.5. Interview summary participant E

The participant explained that, in clinical practice, contours often have worse quality compared to the contours provided during the study. As a result, extensive editing, slice removal, and redrawing are common tasks, and this habit initially influenced her approach in this study. In the clinic, rigid registration is frequently used and is often inaccurate, requiring many slices to be deleted. In contrast, the contours provided in this study were of high quality. At first, the participant approached them with the same mentality of correcting every millimetre, but after working on several patients, they realised this level of modification was unnecessary. Many of the changes were minor, did not affect the CTV, and took considerable time. This shift in mindset occurred during session 2, although possibly

already toward the end of session 1.

The participant did not perceive a difference in workload between the two sessions. The main difference was that in the second session, they felt more confident that the automatic contour was correct, and corrections were small. Consequently, editing was expected to take less time.

At the beginning, the participant used the uncertainty map from the sagittal view. The workflow involved first assessing the provided contour, then checking the map for magenta areas. They switched the map on when they felt uncertain, but also when they felt confident, mainly out of curiosity to see whether the map indicated magenta regions. The participant noted that the areas they edited were sometimes magenta and sometimes green; therefore, they did not consistently agree with the map. Most of the time, the participant felt very certain about their edits; only in one or two cases they did experience doubt, and in those instances, the map was helpful. In the remaining cases, they already had high certainty.

For difficult cases, the participant expressed a preference for having both the reference and the map available. However, they reported limited trust in the map because some areas that seemed correct appeared magenta, and some areas they considered incorrect appeared green. In one case, the participant missed a lymph node. They said that such oversights can occur and were not influenced by the fact that the map indicated green in that region. They noted that recognition can differ depending on the moment.

The participant found the map visually thick, making it difficult to see the underlying anatomy, requiring frequent switching off. Typically, they examined the sagittal view and then switched off the map when delineating slice by slice. The map was activated mainly when in doubt. Furthermore, the participant tended to zoom in more when dealing with uncertainty or difficult cases.

The participant mentioned that editing behaviour depends on the day and on the mindset. If they began in an "editing mode" from the lower portion of a structure, they tended to continue extensively, explaining that "if you change something, you want to do it well, so you keep going". Moreover, the participant recalled spending more or less time on certain parts of structures, but when reviewing the video, it became clear that their perception did not match what actually happened. The participant also expressed surprise at the intra-observer variability.

For some patients, the participant remembered the anatomy. Moreover, the participant reported that the map did not influence the time required and the confidence. They stated that the map would be acceptable to use but would likely not be used frequently because there is not a strong need for it. For visualisation, it was suggested merging the contour and the map into a single coloured contour that could be toggled on or off together.

When asked whether increased experience with the map could change its usefulness, the participant responded that this would depend on the level of trust developed over time. Currently, there is no trust in the uncertainty estimation, but if future experience shows it to be accurate, they might give more attention and value to it. At present, they trusted their own judgement more.

The participant declared to be one of the most recent RTTs to begin working on rectal cases, having started approximately six months before the interview. The two-month interval between sessions was sufficient for a learning curve and substantial changes in mindset

and editing approach. Finally, the participant reported being "shocked" upon discovering that the maps represented inter-observer variability.

### E.4.6. Interview summary participant F

The participant reported disliking the need to continuously switch the map on and off, as the overlay made it difficult to see fine anatomical borders. Because the map obscured the underlying anatomy, its usability was limited. Overall, the participant did not see a clear added value in using the map.

The contours in the study were perceived as higher quality than those typically obtained in clinical practice. Moreover, according to the participant, given the lack of additional anatomical information, they did not add or delete cranial or caudal slices, reflecting a change in mindset compared to their usual clinical approach. Regarding anatomical familiarity, the participant initially stated they did not remember the anatomy because many patients had been treated in between; later, they mentioned remembering the anatomy for one particular patient. The participant also indicated they did not discuss the cases with colleagues after session 1.

At several points, the participant responded with "I don't know", indicating limited awareness of how or whether the maps influenced the editing behaviour. The participant also noted that they did not use the map very often and that differences observed between the two sessions were likely due to day-to-day variability, including differences in attention and overall mindset. Later, the participant stated that the green areas on the map influenced them: when they saw green, they tended to follow the provided contour more closely and trusted those regions. In contrast, when the map showed magenta areas, they would have preferred having additional proposed contours. The magenta areas did not provide useful information because the participant already knew which areas were uncertain.

Furthermore, the participant initially said confidence was the same with and without the map, but later added that they felt slightly more confident when the map was available, since they trusted the green regions. In clinical practice, the participant usually has a reference planning contour provided by the doctors and always discusses the case with them before or after the first fraction; the doctor also indicates specific areas requiring special attention. This differs from the experimental setup.

The participant sometimes could not explain why edits differed between sessions. They also mentioned that the colourmap intensity influenced the perception; brighter colours would have made the map more noticeable and therefore more influential, though this could worsen anatomical visibility. The participant reported no difference in workload between March and June. The participant reported being experienced with Mirada, using it for image post-processing. According to the participant, the cases in this study were simple, largely due to the presence of substantial fat tissue. In clinical practice, cases are not always so clear. For more challenging cases (e.g., patients with less fat), the participant believed that the map might be more helpful. Finally, the participant reported using the map more as a confirmation tool rather than as a primary guide. They felt the map could be more beneficial for doctors, potentially saving them time, while being less helpful during an online workflow for RTTs.

