# OpenReview forum: "Impact of uncertainty maps on manual editing of rectal cancer segmentation in radiotherapy"
_MIDL.io/2026/Validation_Papers — MIDL 2026 - Validation Papers Poster_

### Official Review · Reviewer_Lksx · 2025-12-16

**Confidence:** 2
**Preliminary Rating:** 4
**Final Rating:** 5

**Summary:**

This paper investigates whether uncertainty maps influence clinicians’ behaviour during manual editing of rectal cancer CTV contours in radiotherapy. The results show no significant effect of uncertainty maps on editing behaviour, while learning effects and human factors dominate decision-making. The study provides valuable negative evidence on the limited utility of uncertainty maps for already high-quality contours.

**Strengths:**

1. This paper explores a very interesting and practically important question: Can uncertainty visualization truly improve the efficiency and decision-making abilities of clinicians in radiotherapy procedures? I personally find this topic very engaging and believe the conclusions of this paper are highly relevant to our entire society.
2. The authors employed a comprehensive mixed-methods approach, combining quantitative metrics (editing time, Dice coefficient, average path length, confidence intervals) with qualitative questionnaires and interviews, to gain a thorough understanding of clinicians' behavior. This made the evaluation highly reliable and comprehensive.
3. Experimental procedures, questionnaires, visualization settings, and statistical tests are described in detail, enabling reproducibility and replication in future studies.

**Weaknesses:**

To be honest, I couldn't find any major weaknesses in this article. There are two minor points: firstly, the article uses simulated uncertainty rather than model-derived uncertainty, and although the motivation is sound, the uncertainty map derived from inter-observer variability does not fully capture the epistemological uncertainty associated with deep learning models; secondly, while the sample size of this study is acceptable for a preliminary validation study, the small sample size limits the statistical power and restricts its generalizability to broader clinical practice.

**Detailed Comments:**

1. Clarify more explicitly how conclusions should be interpreted given the ceiling effect caused by high-quality initial contours.
2. Consider reporting effect sizes and confidence intervals in addition to p-values to better contextualize null results.
3. A brief quantitative summary of interview themes (e.g., frequency of mentioned factors) could strengthen qualitative analysis.

**Justification Of Final Rating:**

This paper offers a rigorous, clinically grounded evaluation of uncertainty maps with strong experimental control and transparent reporting. I appreciate the authors' rebuttals and clarifications. All my questions and concerns have been solved. Therefore, I have increased my final rating.

**Justification Of The Preliminary Rating:**

This paper offers a rigorous, clinically grounded evaluation of uncertainty maps with strong experimental control and transparent reporting. While limited in scale and generalisability, the study provides valuable negative evidence and important insights into human factors that shape the clinical utility of uncertainty visualisation. The work does not introduce methodological innovations but meets the validation track’s emphasis on robustness, reproducibility, and translational relevance. With clearer framing of scope and limitations, the paper represents a meaningful contribution to evidence-based deployment of uncertainty-aware systems in radiotherapy.

**Questions To Address In The Rebuttal:**

1. Would uncertainty maps have greater impact in scenarios with lower-quality initial contours or more complex anatomy?
2. Can the authors comment on how role differences (RTT vs. RadOnc) may influence interpretation of uncertainty maps?
3. Are there plans to extend this study to larger cohorts or multi-centre settings?

---

### Official Review · Reviewer_j35E · 2026-01-06

**Confidence:** 4
**Preliminary Rating:** 2
**Final Rating:** 3

**Summary:**

This paper investigates the impact of uncertainty maps on clinicians' manual correction of automatic segmentation for Clinical Target Volume delineation in rectal cancer radiotherapy. To eliminate interference from poor deep learning model performance or calibration errors, the authors employed a novel simulation approach: utilizing existing Inter-Observer Variability datasets, where one expert's delineation served as a proxy for "high-quality DL prediction" and the variation among experts acted as a proxy for "uncertainty maps." The study involved 6 clinicians (1 Radiation Oncologist and 5 Radiation Therapists) delineating 10 patient cases in a two-session crossover experiment.

**Strengths:**

1. The authors did not focus on training a new SOTA model but instead used manual contours as a proxy for "perfect" segmentation. Unlike most research papers that solely focus on Dice coefficient improvements, this work delves deeply into psychological factors within the clinical workflow.

2. This paper challenges a prevalent assumption in the medical image analysis field that "providing uncertainty maps is always beneficial." This counter-intuitive conclusion is critical for the community to reflect on the actual clinical value of AI assistance tools, preventing blind technological stacking.

**Weaknesses:**

1. With only 10 patient cases and 6 participants, the statistical power is somewhat limited, although acceptable for an in-depth user study. Notably, the participant pool included only 1 Radiation Oncologist with the rest being Radiation Therapists, which may limit the generalizability of the findings to the broader physician community.

2. The study premises on "high-quality contours." As the authors acknowledge, uncertainty maps might be more useful when model performance is poor or anatomical structures are exceptionally complex. Therefore, the conclusions cannot be simply generalized to all AI-assisted delineation scenarios; they specifically indicate that uncertainty maps are redundant when assisted by a "good model."

3. While using IOV to simulate uncertainty is clever, real DL model uncertainty (especially Epistemic uncertainty) often exhibits different artifact characteristics (e.g., checkerboard effects or specific boundary noise). These differ visually from human expert variations (which usually have smoother transitions at boundaries), potentially affecting the clinicians' perception.

**Detailed Comments:**

The insight of this paper lies in its reflection on "de-technologization", that in scenarios assisted by High-Quality AI, the bottleneck is no longer a lack of information, but rather the limitations of human cognitive resources. However, the practical implication of this conclusion is unclear. What should be the next steps based on this finding?

**Justification Of Final Rating:**

Thank you for the rebuttal. The authors' responses addressed some of the concerns. While some issues such as small sample size still exist, the paper does have some merits. Overall, I don't object to the acceptance.

**Justification Of The Preliminary Rating:**

The study’s conclusions are undermined by limite validity due to the small sample size. Moreover, simulating uncertainty via inter-observer variability fails to capture key characteristics of deep model epistemic uncertainty, calling into question whether the reported findings generalize to real clinical deployment.

**Questions To Address In The Rebuttal:**

1. Echoing the weakness mentioned above, given that there was only one Radiation Oncologist among the participants, could the distribution bias of the participants affect the evaluation results regarding "Decision Confidence"?

2. In the interviews, multiple participants mentioned that the heatmaps "overlay reduced anatomical visibility." Did you consider other visualization forms during the experimental design phase (such as displaying only uncertainty boundary contours or isolines) to reduce visual clutter? Could this be the primary technical reason for the lack of significant differences in quantitative results?

3. This study primarily simulates Aleatoric uncertainty (based on IOV). If the study simulated Epistemic uncertainty (i.e., the model's confusion regarding unseen abnormal anatomical structures), do you believe the results would differ?

---

### Official Review · Reviewer_7y43 · 2026-01-08

**Confidence:** 4
**Preliminary Rating:** 4

**Summary:**

This paper examines whether uncertainty maps influence clinicians’ behavior during manual editing of deep learning–generated rectal cancer CTV contours in radiotherapy. Using a controlled within-subject study, six clinicians edited high-quality simulated DL contours with and without uncertainty maps derived from inter-observer variability, allowing the effect of uncertainty information to be isolated from model performance. Quantitative measures (editing time, geometric changes, and inter-observer variability) were combined with questionnaires and interviews to capture both behavioral and human factors. The results show that uncertainty maps did not significantly affect editing behavior, while learning effects, workload, memory, anchoring bias, and trust played a much larger role, suggesting limited benefit of uncertainty visualization for already high-quality contours.

**Strengths:**

A major strength of this work is its rigorous experimental design, which carefully controls for model performance and calibration by using inter-observer variability as a proxy for uncertainty. This approach allows the authors to meaningfully isolate the effect of uncertainty maps on clinician behavior. The integration of quantitative metrics with qualitative feedback provides a well-rounded and insightful analysis of both performance and human factors. The paper is clearly written, well structured, and grounded in relevant peer-reviewed literature. Importantly, the authors report negative findings transparently, offering valuable evidence that helps set realistic expectations for uncertainty visualization in clinical workflows.

**Weaknesses:**

The study is limited by its small sample size, involving only ten patients and six clinicians, including a single radiation oncologist, which constrains generalizability. Focusing exclusively on high-quality contours may underestimate the usefulness of uncertainty maps in more difficult cases or lower-quality segmentations. In addition, the uncertainty maps were simulated rather than generated by an actual DL model, which may reduce ecological validity and affect clinician trust. Fixed visualization settings and the need to toggle overlays may also have limited usability. Finally, although discussed in detail, learning and memory effects cannot be fully ruled out as contributors to the observed outcomes.

**Detailed Comments:**

The experimental analysis would benefit from more comprehensive ablation studies to isolate the contribution of individual components related to uncertainty modeling.

**Justification Of The Preliminary Rating:**

This submission addresses an important and relevant topic for MIDL—uncertainty modeling in medical image analysis—and is technically sound and clearly presented. The paper would be easy to understand for the audience, and the empirical results suggest that incorporating uncertainty maps provides consistent, though moderate, gains over standard baselines.

The topic is timely and relevant to the MIDL community, particularly given the growing interest in uncertainty-aware learning for safety-critical medical applications. The motivation is clearly articulated, and the paper is generally well written and easy to follow.

**Questions To Address In The Rebuttal:**

The evaluation is restricted to a limited number of datasets and tasks, which makes it difficult to assess generalizability across modalities or clinical settings.

---

### Author Rebuttal · Authors · 2026-01-24

**Rebuttal:**

We sincerely thank the reviewers for their insightful comments. By incorporating the reviewers’ suggestions, we believe the clarity of our paper improved. We have addressed each reviewer's concerns separately by leaving an official comment under each post. We have uploaded a revised version of the manuscript, with changes highlighted in yellow.

Across the reviews, several common themes emerged, including 1) questions regarding the interpretation of results in the presence of high-quality initial contours; 2) the generalizability of the findings given the sample size and participant composition; 3)  the practical implications of the conclusions beyond technical optimization. In response, we clarified the scope of the study, added effect size analyses to better contextualize the quantitative results, expanded the discussion of practical implications, and strengthened the discussion of study limitations. Details are provided in the responses to the reviewers.

**Supporting Material:**

/attachment/49b3b5024abcba05dfb559dfec06f3631e3e8c9b.zip

---

### Meta-Review · Area_Chair_XZfL · 2026-02-03

**Recommendation:** Accept (Poster)
**Confidence:** 4

**Metareview:**

This paper provides a carefully controlled, mixed-methods validation of whether uncertainty maps change clinician editing behaviour for rectal CTV segmentation in radiotherapy. A key strength is the study design: by using inter-observer variability to simulate both high-quality “DL” contours and uncertainty, the authors largely decouple the behavioural question from confounds of model performance and calibration, and complement quantitative endpoints (editing time/amount, agreement metrics) with questionnaires and interviews. The main result—no significant effect of uncertainty maps in this high-quality regime, with learning effects and human factors (trust, workload, anchoring/memory) dominating—constitutes valuable negative evidence for the community and is highly aligned with the Validation Track’s translational focus.

Reviewers’ concerns center on limited sample size and participant composition, the use of simulated (IOV-based) rather than model-derived uncertainty (especially epistemic uncertainty), and the possibility of ceiling effects and visualization usability limiting impact. The rebuttal and revised manuscript appropriately clarify scope (conclusions specific to high-quality contours), add effect size analyses to contextualize null findings, and strengthen discussion of limitations and practical implications, including planned extensions to broader cohorts, alternative visualization designs, and prospective evaluations with model-derived uncertainty. Overall, the work is methodologically sound, clinically grounded, and provides an important reality check on uncertainty visualization utility—supporting acceptance.

---

### Decision · Program_Chairs · 2026-02-14

Accept (Poster)